# Sec17/Sec18 can support membrane fusion without help from completion of SNARE zippering

Hongki Song[1†], Thomas L Torng[1†], Amy S Orr[1], Axel T Brunger[2], William T Wickner[1]*

[1]Department of Biochemistry and Cell Biology Geisel School of Medicine at Dartmouth, Hanover, United States; [2]Howard Hughes Medical Institute and Department of Molecular and Cellular Physiology Stanford University, Stanford, United States

**Abstract** Membrane fusion requires R-, Qa-, Qb-, and Qc-family SNAREs that zipper into RQaQbQc coiled coils, driven by the sequestration of apolar amino acids. Zippering has been thought to provide all the force driving fusion. Sec17/αSNAP can form an oligomeric assembly with SNAREs with the Sec17 C-terminus bound to Sec18/NSF, the central region bound to SNAREs, and a crucial apolar loop near the N-terminus poised to insert into membranes. We now report that Sec17 and Sec18 can drive robust fusion without requiring zippering completion. Zippering-driven fusion is blocked by deleting the C-terminal quarter of any Q-SNARE domain or by replacing the apolar amino acids of the Qa-SNARE that face the center of the 4-SNARE coiled coils with polar residues. These blocks, singly or combined, are bypassed by Sec17 and Sec18, and SNARE-dependent fusion is restored without help from completing zippering.

*For correspondence: William.T.Wickner@dartmouth.edu

[†]HS and TLT are co-first authors of this work

## Introduction

Membrane fusion requires Rab-family GTPases and SNARE proteins. SNAREs constitute four families, termed R, Qa, Qb, and Qc (*Fasshauer et al., 1998*). Each of them has an N-domain, an α-helical SNARE domain of 50–60 aminoacyl residues with heptad-repeat apolar residues, and often a C-terminal membrane anchor. Each SNARE α-helical turn is termed a 'layer.' The central '0-layer' of each fully-assembled SNARE complex has inwardly-oriented arginyl (for R-SNAREs) or glutaminyl (for Qa, Qb, and Qc SNAREs) residues, forming a polar center to the otherwise hydrophobic core of the 4-helical SNARE bundle (*Sutton et al., 1998*). The SNARE domain layers are numbered from the 0-layer, in the positive direction toward the SNARE C-termini and in the negative direction toward the N-domains. Prior to 4-SNARE assembly, individual SNARE domains are random coil (*Fasshauer et al., 1997*; *Hazzard et al., 1999*). Sec1/Munc18 (SM) family proteins catalyze the N- to C-directional assembly of SNAREs anchored to each tethered membrane (*Fiebig et al., 1999*; *Sørensen et al., 2006*; *Baker et al., 2015*; *Orr et al., 2017*; *Jiao et al., 2018*). Each SNARE domain transitions from random coil to α-helix as the heptad-repeat apolar amino acyl residues become sequestered into the interior of the coiled coils (*Fasshauer et al., 1997*; *Sutton et al., 1998*). This hydrophobic collapse relies on the exclusion of water and is the driving force for SNARE assembly (*Sørensen et al., 2006*). Completion of SNARE zippering can release up to 40 kBT per SNARE complex (*Gao et al., 2012*; *Min et al., 2013*; *Zhang, 2017*) to overcome the 40–90 kBT hydration barrier for membrane stalk formation, the dominant energy barrier for fusion (*Aeffner et al., 2012*). Upon fusion, the *trans*-SNARE complex becomes a *cis*-complex, anchored to the fused membrane bilayer. Sec17 (αSNAP) and SNAREs are receptors for the Sec18 (NSF) AAA ATPase (*Clary et al., 1990*; *Winter et al., 2009*; *Zick et al., 2015*). Sec18 uses the energy from ATP binding and hydrolysis to

disassemble SNAREs for further fusion cycles (*Söllner et al., 1993*; *Ungermann et al., 1998*; *Zhao et al., 2015*) and to disassemble dead-end SNARE complexes (*Xu et al., 2010*; *Lai et al., 2017*; *Choi et al., 2018*; *Song and Wickner, 2019*; *Jun and Wickner, 2019*).

The molecular interactions between Sec18/NSF, Sec17/αSNAP, and neuronal SNAREs were illuminated by determination of their structures when assembled without membrane anchors into the NSF/αSNAP/SNARE complex, also referred to as the 20S particle (*Zhao et al., 2015*; *White et al., 2018*). The heart of these structures is the 4-helical bundle of the R, Qa, Qb, and Qc SNARE domains. Between two and four Sec17/αSNAP form a right-handed assembly surrounding the left-handed superhelical coiled coils of the SNARE complex. In this structure, the N-terminal apolar loop of each αSNAP is poised to enter a lipid bilayer adjacent to the SNARE transmembrane (TM) domains.

Yeast vacuole fusion, a model of non-neuronal fusion, has been studied in vivo (*Wada et al., 1992*), in vitro with the isolated organelle (*Wickner, 2010*), and in a reconstituted proteoliposome-based reaction with purified components (*Mima et al., 2008*; *Stroupe et al., 2009*; *Zick and Wickner, 2016*). Each protein implicated by the in vivo genetics is required for the reconstitution: the Rab Ypt7, the R-SNARE Nyv1, and Q-SNAREs Vam3, Vti1, and Vam7 (hereafter referred to as R, Qa, Qb, and Qc), and a large hexameric protein termed HOPS (**ho**motypic fusion and vacuole **p**rotein **s**orting) with multiple direct affinities. Two HOPS subunits bind Ypt7, anchored on each membrane (*Brett et al., 2008*), to mediate tethering (*Hickey and Wickner, 2010*). A third HOPS subunit is Vps33, the vacuolar Sec1/Munc18 (SM) protein, with direct capacity to bind R and Qa SNARE domains, in parallel and in register (*Baker et al., 2015*). HOPS also has direct affinity for the Qb and Qc SNAREs (*Stroupe et al., 2006*; *Song et al., 2020*) and for vacuolar lipids (*Orr et al., 2015*). Some functions of HOPS correspond to fusion factors in other systems; neuronal Munc13 cooperates with Munc18 in SNARE assembly (*Richmond et al., 2001*; *Ma et al., 2011*; *Lai et al., 2017*). Munc18 corresponds to the HOPS subunit Vps33, but the protein that mediates the Munc13 function for vacuole fusion is unclear. The association of HOPS with Ypt7 and vacuolar lipids allosterically activates HOPS to catalyze SNARE assembly (*Torng et al., 2020*). When the SNAREs are initially in 4-SNARE complexes on two apposed membranes, fusion requires Sec17, Sec18, and ATP to disassemble these *cis*-SNARE complexes and liberate the SNAREs for assembly into *trans*-complexes (*Mayer et al., 1996*; *Nichols et al., 1997*; *Zick et al., 2015*).

Though early studies have suggested that disassembly of *cis*-SNARE complexes might be the sole function of Sec17 and Sec18 (*Mayer et al., 1996*), recent findings have broadened our understanding of their roles. While SNAREs are a core fusion machine (*Weber et al., 1998*), a complete fusion machine (*Mima et al., 2008*; *Zick and Wickner, 2016*) also requires the Rab, its tethering effector, an SM-family protein, and the NSF/Sec18 and αSNAP/Sec17 SNARE chaperones. In this more complete context, Sec17 and Sec18 also contribute to fusion per se: (1) *trans*-SNARE complexes, which form in a Ypt7-dependent manner between vacuoles, bear Sec17 in comparable abundance to the SNAREs (*Xu et al., 2010*). (2) Fusion between proteoliposomes has been reconstituted with purified components. When tethering is by non-specific agents, fusion is inhibited by Sec17, Sec18, and ATP (*Mima et al., 2008*; *Song et al., 2017*). However, Sec17, Sec18, and ATP stimulate fusion with HOPS (*Mima et al., 2008*; *Zick et al., 2015*; *Song et al., 2017*). (3) A pioneering study by *Schwartz and Merz, 2009*, showed that the Qc3Δ deletion of several heptads at the C-terminus of the Qc SNARE blocks the fusion of isolated yeast vacuoles, but this block is overcome by the addition of Sec17. Qc3Δ, ending at the SNARE domain layer +3 and thus lacking layers +4 to +8, blocks vacuole fusion in vivo as well, and overexpression of Sec17 partially restores cellular vacuole morphology (*Schwartz et al., 2017*). Reconstituted in vitro fusion with limiting Sec17 concentrations, where Sec17 will not restore fusion, also requires Sec18 (*Schwartz et al., 2017*). It has been unclear whether the Sec17 bypass of this deletion of the C-terminal Qc region is particular to just this one SNARE or is general for any Q-SNARE, and whether Sec17 simply contributes its SNARE-binding energy to the energy of 3-SNARE zippering or whether it drives fusion by other means. (4) Fusion reactions with initially separate SNAREs can require Sec17, and this fusion is stimulated by Sec18 without ATP hydrolysis (*Zick et al., 2015*; *Song et al., 2017*). Sec17 alone stimulates the fusion of reconstituted proteoliposomes with wild-type SNAREs, and the degree of stimulation is a function of the lipid headgroup and fatty acyl composition (*Zick et al., 2015*). An intermediate in fusion accumulates during HOPS-dependent fusion without Sec17, allowing a sudden burst of fusion upon Sec17 addition (*Zick et al., 2015*). Single-molecule pulling studies have also revealed that αSNAP stabilizes

SNARE complexes (*Ma et al., 2016*, but see *Ryu et al., 2015*). However, the mechanism by which HOPS-dependent fusion is stimulated by Sec17/Sec18 has been unclear.

Complete SNARE zippering is considered essential for SNARE-dependent fusion (*Sørensen et al., 2006*). We now report that Sec17 has a second mode of promoting fusion, which can compensate for incomplete zippering. Fusion with SNAREs and HOPS is completely arrested when several heptad repeats in the C-terminal region of the SNARE core complex are removed from any one of the three Q-SNAREs. With any such C-terminally truncated Q-SNARE domain, or even with C-terminal truncations to both Qb and Qc, blocked fusion is restored by Sec17 and Sec18 without ATP hydrolysis. Association between the C-terminal heptads of the R and the single remaining full-length Qa-SNARE, each anchored to one of the docked membranes, would not yield the same assembly energy as with wild-type SNAREs (*Sutton et al., 1998*) and would contribute far less force toward the bilayer rearrangements of fusion. The N-terminal apolar loop of Sec17 is particularly important for this function of Sec17, and it may stabilize SNARE bundles or trigger fusion by insertion into lipid bilayers. Zippering-driven fusion is also arrested with full-length SNARE domains when the apolar, inward-facing residues of the Qa SNARE layers +4 to +8 are replaced by Ala, Ser, or Gly, but in each case fusion is restored by Sec17, Sec18, and non-hydrolyzable ATPγS. Strikingly, even fusion that is blocked by the concurrent replacement of apolar residues from the +4 to +8 layers of Qa and the deletion of the +4 to +8 layers of both Qb and Qc, removing all capacity for hydrophobic collapse between the +4 and +8 layers of the R and Q SNAREs, is fully restored by Sec17 and Sec18. We propose that Sec17 either creates a favorable folding environment for the assembly of the remaining full-length SNARE domains or directly promotes bilayer remodeling through insertion of the apolar loops of several SNARE-bound Sec17s or acts by a combination of these two mechanisms.

## Results

Vacuole SNAREs (*Figure 1A*) have an N-domain and a SNARE domain. Several of them have a TM anchor, but Qc lacks a hydrophobic membrane anchor, and instead associates with membranes through its affinities for the other SNAREs, for HOPS (*Stroupe et al., 2006*), and for phosphatidylinositol 3-phosphate through its N-terminal PX domain (*Cheever et al., 2001*). SNARE domain layers are numbered from the central 0-layer, as shown. Fusion requires that R- and at least one Q-SNARE be anchored to docked membranes (*Song and Wickner, 2017*). When they are, soluble forms of the other Q-SNAREs without membrane anchors, termed sQ (*Figure 1A*), will support Ypt7/HOPS-dependent fusion. Vacuolar SNAREs with C-terminal truncations, corresponding to partial zippering, form stable complexes (*Figure 1—figure supplement 1*), supporting their use in fusion studies. Based on the single-particle cryo-EM (Electron Microscope) structure of the homologous human neuronal NSF/αSNAP/SNARE complex (*Zhao et al., 2015*), we modeled the associations of Sec17 (αSNAP) (*Figure 1B*) and Sec18 (NSF) with vacuolar SNAREs, viewed in profile or in an end-on view from the membrane (*Figure 1C*). In this model, we assume that four Sec17 monomers (*Figure 1B*) assemble together, surrounding the 4-SNARE coiled coil (*Figure 1C*). In contrast, only two αSNAP molecules have been observed in EM structures of the NSF/αSNAP/V7-SNARE complex (*Zhao et al., 2015*) and the NSF/αSNAP/SNARE complex that included the linker between the two SNAP-25 SNARE domains (*White et al., 2018*). The presence of the SNAP-25 linker in these two complexes may interfere with the binding of the other two αSNAP molecules. Since the vacuolar SNARE complex does not contain a linker between SNARE domains, it is reasonable to postulate that four Sec17 molecules bind to the vacuolar SNARE complex, but the precise number of Sec17 molecules is yet to be determined for the Sec18/Sec17/vacuolar SNARE complex. Rapid fusion needs Sec17 and Sec18 in addition to HOPS and SNAREs (*Song et al., 2017*). To understand how they work together, we exploited direct assays of SNARE associations to show that incompletely zippered SNAREs can associate stably and that HOPS allows Sec17 to promote the completion of zippering and SNARE complex stability. We then examined their functional relationships to show that SNARE-bound Sec17 and Sec18 can promote rapid fusion when energy from zippering is greatly reduced or lost.

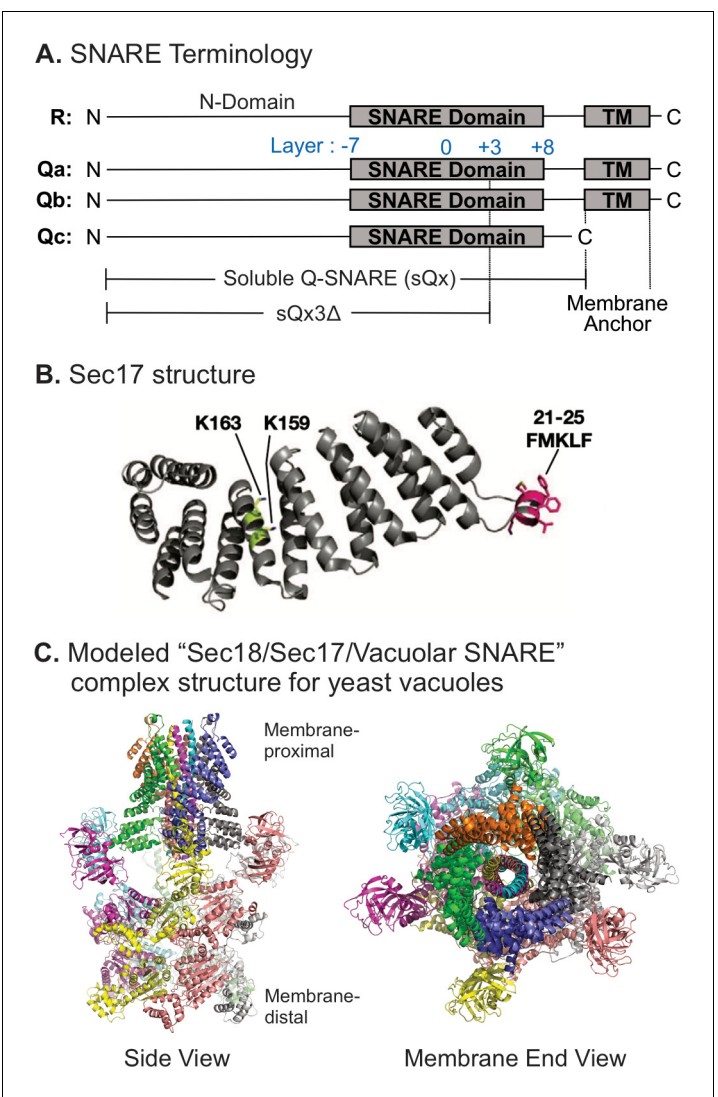

**Figure 1.** Model of the Sec18/Sec17/vacuolar SNARE complex and Sec17 mutations. (**A**) Schematic of the four yeast vacuolar SNAREs, the soluble Q-SNAREs (sQx), and their deletion derivatives sQx3Δ lacking regions C-terminal to the +3 layer of the SNARE domain. (**B**) Structure of Sec17 (***Rice and Brunger, 1999***, PDB ID 1QQE). Residues mutated in certain experiments are highlighted as in ***Schwartz et al., 2017***. (**C**) Modeling of the vacuolar Sec18/Sec17/SNARE complex. MODELLER (***Webb and Sali, 2016***) was used to create individual homology models of the vacuolar SNARE complex (Nyv1, Vam3, Vti1, Vam7) and of Sec18 starting from the coordinates of synaptobrevin-2, SNAP-25, syntaxin-1A, and NSF in the cryo-EM structure of the neuronal 20S complex (PDB ID 3J96) (***Zhao et al., 2015***). These homology models, together with the crystal structure of Sec17 (PDB ID 1QQE) (***Rice and Brunger, 1999***), were fit into the cryo-EM structure of the neuronal 20S complex (PDB ID 3J96) (***Zhao et al., 2015***). We used PDB ID 3J96 because this cryo-EM structure did not include the SNAP-25 linker and the $H_{abc}$ domain of syntaxin-1A. The vacuolar SNARE complex (Nyv1, Vam3, Vti1, Vam7) also does not include a linker between any of the SNARE motifs; in all structures of NSF/αSNAP/ternary SNARE complexes determined thus far, four αSNAP molecules are observed for SNARE complexes that do not contain a linker connecting two of the SNARE domains (***White et al., 2018***), and we therefore included four Sec17 molecules in our model of the vacuolar 20S complex. In the PDB coordinate file supplied as Source Data File of the homology model of the Sec18/Sec17/vacuolar SNARE complex, Sec18 molecules, chains A–F; Sec17, chains G–J; Nyv1, chain K; Vam3, chain L; Vti1, chain M; Vam7, chain N are included. Colors: cyan: Nyv1; magenta: Vam3; yellow: Vti1; salmon: Vam7; gray, orange, green, slate: Sec17; yellow, magenta, gray, blue, salmon, green: Sec18. Cartoon representations are shown. Two views related by a 90-degree rotation are included (left: side view; right: membrane-end view).

*Figure 1 continued on next page*

*Figure 1 continued*

The online version of this article includes the following source data and figure supplement(s) for figure 1:

**Source data 1.** Source data file (PDB) for *Figure 1C*.

**Figure supplement 1.** SNARE associations during partial zippering.

## Sec17 alters SNARE complex conformation

The capacity of vacuolar SNAREs to form stable partially zippered structures raised the question of whether these SNAREs can zipper efficiently, especially when anchored to membranes or associated with other fusion proteins such as HOPS. To study the kinetics of SNARE interactions, we employed an ensemble fluorescence resonance energy transfer (FRET) assay of vacuolar SNARE associations (*Torng et al., 2020*). The Qc-SNARE was prepared with a unique cysteinyl residue in any of three positions (*Figure 2A*), either the native Cys208, which is upstream (U) of the SNARE domain or, after substitution of serine for this cysteine, with Met250Cys at the N-terminal end of the SNARE domain or with Ala316Cys at the C-terminal end of the SNARE domain. Each was derivatized with Oregon Green 488. Fusion proteins were also prepared with a unique cysteine either immediately N-terminal, or C-terminal, to the Qb SNARE domain, and each was derivatized with Alexa Fluor 568. As a model for exploring the effects of HOPS and Sec17 on SNARE dynamics, these fluorescent proteins were co-incubated with proteoliposomes bearing Ypt7, R, and Qa. *cis*-SNARE complex assembly can occur spontaneously on these proteoliposomes, but assembly is stimulated by HOPS, allowing direct comparison of HOPS-dependent and HOPS-independent SNARE assembly (*Torng et al., 2020*). The average FRET efficiency in these studies is modest because they include, in bulk reactions, all the fluorescent Qb and Qc, many of which do not enter SNARE complexes. SNARE complex assembly with HOPS gave a high average FRET efficiency when fluorophores were at the N-terminus of the Qb SNARE domain and at or near the N-terminal end of the Qc SNARE domain (*Figure 2B*, red and yellow curves). There was a lower FRET efficiency when the fluorophores were at opposite ends of the Qc and Qb SNARE domains (*Figure 2B*; blue, green, and indigo). When both fluorophores were at the C-terminal ends of the Qb and Qc SNARE domains, a low signal was seen (*Figure 2B*, purple), similar to the average FRET efficiency when fluorophores were at opposite ends of the SNARE domains, thus suggesting incomplete zippering. After 1 hr, Sec17 was added to each reaction. Strikingly, Sec17 only enhanced the average FRET efficiency between C-terminal fluorophores, rapidly rising to the level seen when the fluorophores were together at the N-terminii of the SNARE domains (*Figure 2B*, purple; *Figure 2—figure supplement 1*), indicating a Sec17-induced change at the C-terminal end of the SNARE complex. Sec17/αSNAP may promote the zippering of isolated SNARE domains (*Ma et al., 2016*) but is now seen to act in the context of membranes and HOPS.

## Sec17-induced conformational change requires HOPS

Proteoliposomes with R, Qa, and Ypt7 (where indicated) were incubated with Qb-SNARE domain and Qc. Both SNARE domains were either fluorescently labeled at their N-termini (*Figure 2C*, bars 1–5) or at their C-termini (*Figure 2C*, bars 6–10). Incubations were in the presence or absence of HOPS. Sec17 addition after 1 hr did not enhance the average FRET efficiency between N-terminally disposed fluorophores in the presence or absence of HOPS (*Figure 2C*, bars 1–5), but stimulated the average FRET efficiency between SNARE domain C-terminal fluorophores in a HOPS-dependent manner (*Figure 2C*, bars 6–10), since the enhanced FRET between the SNAREs in the presence of HOPS and Sec17 (bar 9) is not seen without HOPS (bar 8) or without Sec17 (bar 10). This indicates a HOPS-dependent and Sec17-induced SNARE conformational change. This was diminished when zippering was inhibited by the absence of the +4 to +8 layers of sQa (*Figure 2D*, bar 1 vs. 3) or by the conversion of each inward-facing apolar residue of the full-length Qa SNARE domain layers +4 to +8 to Gly (*Figure 2E*, bar 1 vs. 3). The F22SM23S mutation of Sec17 (FSMS hereafter), diminishing the hydrophobicity of its N-domain loop (*Song et al., 2017*), reduced the Sec17 capacity for inducing HOPS-dependent conformational change (*Figure 2F*). We also examined the effects of other mutants of Sec17. The K159E,K163E mutation (KEKE hereafter) diminishes Sec17:SNARE association (*Marz et al., 2003*); one of these residues (Sec17 K159) is in a pair (αSNAP K122, K163) that abolishes disassembly of the neuronal SNARE complex by NSF/αSNAP (*Zhao et al., 2015*). The

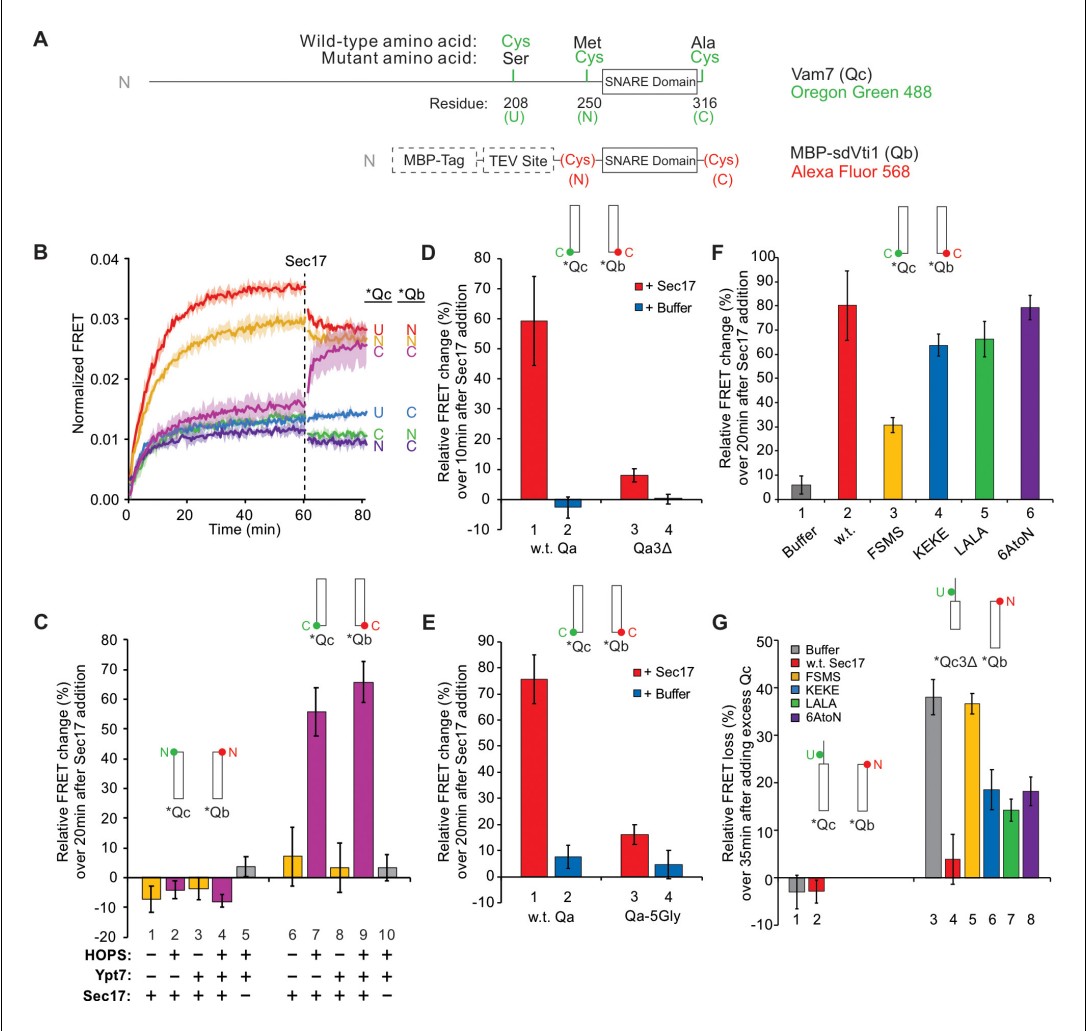

**Figure 2.** Sec17 modifies SNARE conformation in a HOPS- and zippering-dependent manner and stabilizes complexes with truncated SNARE domains. (**A**) Schematic of fluorescently labeled SNARE constructs used in this study. SNAREs were derivatized as described previously (**Torng et al., 2020**). Wild-type Qc contains a single Cys residue at 208 at the upstream (U) position. (N)- and (C)-labeled constructs replace residues 250 and 316 with Cys, while also replacing Cys208 with Ser. Each Qc construct was derivatized with Oregon Green 488 as described (**Torng et al., 2020**). A fusion of maltose-binding protein (MBP), a TEV site, and the Qb SNARE domain (residues 133–187) was expressed with an added cysteinyl residue immediately upstream or downstream of the SNARE domain. Each Qb construct was derivatized with Alexa Fluor 568. Qc and Qb labeled with a fluorescent probe at any position are written as *Qc and *Qb. (**B–G**) Bar graphs are reported as the mean of the relative ensemble fluorescence resonance energy transfer (FRET) change (%) per trial with propagated standard deviation for n = 3 trials. The relative change was calculated by averaging 10 data points, each from just before Sec17 addition and from the end of the measurement period 20 min later, except where indicated. Specific time points used as well as the statistics for the propagation of uncertainty are shown in Supplementary Data. A bar graph representation for (B) and the kinetic curves for (C–G) are provided in **Figure 2—figure supplement 1** . (**B**) Sec17 modifies the conformation of the C-terminus of the SNARE complex. Ypt7/RQa proteoliposomes were incubated with pairs of *Qb and *Qc labeled at the N, C, or upstream (U) locations as indicated in (A). Curves are averages of n = 3 trials, and the shaded regions behind each curve show the standard deviation per time point. (**C**) Sec17-promoted zippering requires HOPS. RQa proteoliposomes were incubated with either the N-labeled *Qb/*Qc pair (left) or the C-labeled *Qb/*Qc pair (right), with Ypt7 and HOPS as indicated. A reaction with a buffer (RB150) added instead of Sec17 serves as a negative control. (**D**) Sec17 does not promote C-terminal zippering if the SNARE domain of Qa is truncated. Ypt7/R proteoliposomes were incubated with C-terminally labeled *Qb and *Qc and either soluble Qa or Qa3Δ, and the relative FRET change was calculated over 10 min after Sec17 or buffer addition. (**E**) Sec17-induced zippering requires the apolar heptad-repeat amino acyl residues in Qa SNARE domain layers +4 to +8. Proteoliposomes bearing Ypt7, R-SNARE, and either wild-type Qa or Qa with the +4 to +8 layers inwardly-oriented apolar residues mutated to Gly were incubated with C-terminally labeled *Qb and *Qc, and then Sec17 or its mutants were added at t = 60 min. (**F**) Sec17-driven SNARE conformational change is stunted by the F22SM23S mutation of Sec17 (FSMS). Ypt/R proteoliposomes were incubated with sQa and C-terminally labeled *Qb and *Qc, and then Sec17 or mutants as indicated were added at t = 60 min. (**G**) Sec17 stabilizes incompletely zippered SNARE complexes. Ypt7/R proteoliposomes were incubated with sQa and C-terminally labeled *Qb and *Qc. Sec17 or the indicated mutants were added at t = 60 min. Non-fluorescent Qc (8.5 μM) was added at t = 80 min, and the loss of FRET over 35 min was measured starting immediately after the addition of non-fluorescent Qc.

*Figure 2 continued on next page*

*Figure 2 continued*

The online version of this article includes the following source data and figure supplement(s) for figure 2:

**Source data 1.** Source data file (Excel) for *Figure 2B*.
**Figure supplement 1.** Statistics and representative kinetic data for *Figure 2*.
**Figure supplement 2.** Direct binding of Sec17 to HOPS.

C-terminal L291A,L292A mutation of Sec17 (LALA hereafter) interferes with its cooperation with Sec18 for SNARE complex disassembly (*Barnard et al., 1997*; *Schwartz and Merz, 2009*; *Zick et al., 2015*), and 6AtoN is the conversion of six inward-facing acidic residues of Sec17, which face basic SNARE residues in the 20S structure (*Figure 1C*), to neutral residues. Neither KEKE, LALA, nor 6AtoN had a large effect on the capacity of Sec17 to promote HOPS-dependent conformational change (*Figure 2F*).

Since this Sec17-induced SNARE conformational change is seen with C-terminal fluors but not with N-terminal fluors, requires HOPS as the SNARE assembly catalyst, and requires SNAREs that can zipper, a major part of the conformational change may be zippering itself. This might be spontaneous after Sec17-induced release of HOPS from SNAREs (*Collins et al., 2005*; *Schwartz et al., 2017*) or be promoted by Sec17 binding along the SNAREs (*Figure 1C*), reflecting its affinity for the C-terminal region of SNARE bundles (*Ma et al., 2016*). HOPS bound to SNAREs may inhibit C-terminal zippering, even though it helps to assemble the N-terminus of the four-helix bundle. Sec17 could then increase FRET because it displaces HOPS. Such C-terminal zippering would be favored by Sec17 binding, but could also be spontaneous in the absence of these factors.

Sec17 also interacts with partially zippered SNARE complex to promote SNARE complex stability. SNARE complex was assembled by HOPS on Ypt7/R proteoliposomes with soluble Qa, with the Qb SNARE domain labeled with a fluorophore at a cysteinyl residue upstream of the SNARE domain, and with Qc of full length (w.t.) or with the 3Δ C-terminal truncation, each bearing a fluorophore at its native cysteinyl residue upstream of the SNARE domain. When the complex of proteoliposomes with these fluorescent Qb and Qc had full-length SNARE domains, it was stable whether or not it included Sec17, as there was no loss of average FRET efficiency after addition of excess non-

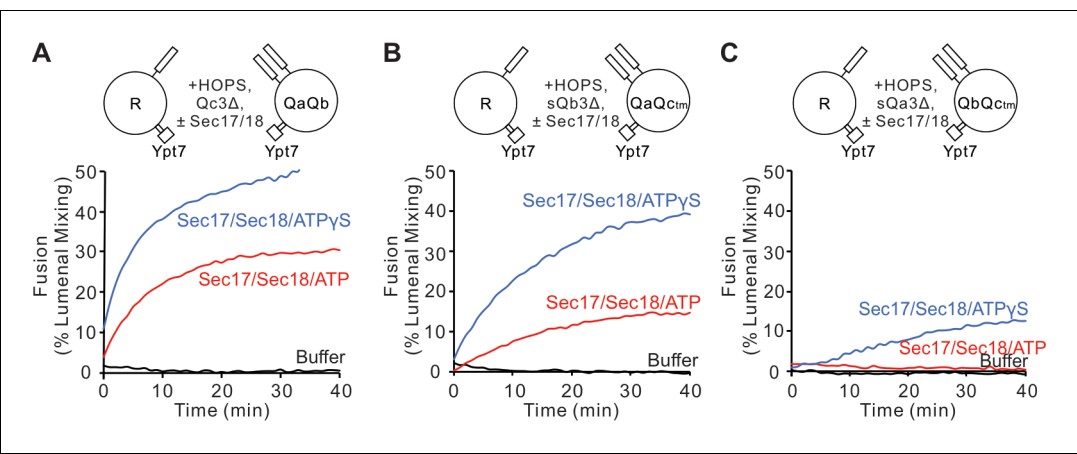

**Figure 3.** Fusion is blocked by deletion of the last five C-terminal SNARE domain layers from any single Q-SNARE and is restored by Sec17, Sec18, and ATP or ATPγS. (**A**) Fusion incubations, as described in 'Materials and methods', had Ypt7/R and Ypt7/QaQb proteoliposomes (1:8000 Ypt7:lipid molar ratio, 1:16,000 SNARE:lipid molar ratio), 2 μM Qc3Δ, and, where indicated, 600 nM Sec17, 300 nM Sec18, 1 mM ATP (red), or ATPγS (blue). (**B**) Fusion with 2 μM sQb3Δ and with Ypt7/QaQc-tm proteoliposomes, but otherwise as in (**A**). (**C**) Fusion with 2 μM sQa3Δ and with Ypt7/QbQc-tm proteoliposomes, but otherwise as in (**A**). Mean and standard deviations from three independent experiments are shown in *Figure 3—figure supplement 1*.

The online version of this article includes the following source data and figure supplement(s) for figure 3:

**Source data 1.** Source data file (Excel) for *Figure 3A,B and C*.
**Figure supplement 1.** Statistics for *Figure 3*.

fluorescent Qc (*Figure 2G*, bars 1 and 2; *Figure 2—figure supplement 1G*). In contrast, fluorescent Qc3Δ was 'chased' by exchange with non-fluorescent Qc (bar 3), but Sec17 stabilized this SNARE complex, blocking the chase (bar 4). Thus, in the absence of Sec17, the assembly of Qc3Δ into partially zippered SNARE complex is reversible. Each domain of Sec17 helps to stabilize Qc3Δ against exchange (bars 5–8), especially the Sec17 N-terminal apolar loop (bar 5). The HOPS-dependent functions of Sec17, such as promotion of zippering, may be aided by the direct affinity between these proteins (*Figure 2—figure supplement 2*).

## Generality of Sec17 and Sec18 bypass of arrested zippering

Sec17 can restore fusion when Qc has truncations at the C-terminal end of its SNARE domain (*Schwartz and Merz, 2009*), stimulated by Sec18 (*Schwartz et al., 2017*). We asked whether Sec17 can restore fusion with similar deletions of residues after the +3 layer in the other Q-SNAREs. Proteoliposomes bearing Ypt7 and R-SNARE were assayed for fusion with proteoliposomes bearing this Rab and any two anchored Q-SNAREs. Proteoliposome mixtures were incubated with HOPS, and the soluble form of the remaining Q-SNARE was deprived of its membrane anchor and a C-terminal portion of its SNARE domain (*Figure 3*; A, Qc3Δ; B, sQb3Δ; C, sQa3Δ). As reported by *Schwartz et al., 2017*, there was no fusion with Qc3Δ unless 600 nM Sec17, 300 nM Sec18, and ATP or ATPγS were present (*Figure 3A*); these concentrations are in the physiological concentration ranges of Sec17 (150–1100 nM) and Sec18 (250–760 nM) (*Ho et al., 2018*). While ATP and its non-hydrolyzable analog ATPγS support comparable fusion with wild-type SNAREs (*Song et al., 2017*), hydrolyzable ATP inhibits fusion through Sec17/Sec18-mediated disassembly of SNARE complexes when defective SNAREs such as Qc3Δ are present, a proofreading function. The same pattern was seen for fusion with sQb3Δ (*Figure 3B*) and sQa3Δ (*Figure 3C*). The unique spatial disposition of the Sec17/αSNAP molecules with respect to each SNARE (*Figure 1C* and *Zhao et al., 2015*) makes it unlikely that Sec17 could somehow fill each of the gaps left by each of these deletions to shield apolar residues within the SNARE bundle and thereby continue to drive zippering, or that Sec17 binding could induce the remaining R and two Q +4 to +8 layers to somehow rotate to form a hydrophobic two- or three-layered core.

Fusion assays were also performed with Ypt7/R proteoliposomes and each of the three Ypt7/single-anchored Q-SNARE proteoliposomes in the presence of HOPS and the other two soluble Q-SNAREs (*Figure 4*). With membrane-anchored Qa and with sQb and Qc having complete SNARE domains, there was HOPS-dependent fusion without further addition (*Figure 4A*, black line), though Sec17 and Sec18 with AMP-PNP, ATPγS, or ATP did stimulate (compare black curves, A–D). Deletion of the +4 to +8 layers from either soluble Qb or Qc abolished fusion (*Figure 4A*), which was restored by Sec17 and Sec18 with either AMP-PNP, ATPγS, or ATP (B–D, red and blue curves). There was no fusion when both soluble Q-SNAREs had truncated SNARE domains (*Figure 4A*, orange), but, strikingly, the fusion was partially restored by Sec17 and Sec18 with ATP (*Figure 4D*, orange) and more fully restored when the adenine nucleotide was resistant to hydrolysis (*Figure 4*, B and C, orange). With two Q-SNAREs lacking the C-terminal portion of their SNARE domains, the apolar amino acyl residues of the remaining two SNAREs would not be as effectively shielded from water if they continued zippering together. Fusion could not be restored by Sec17 and Sec18 if either soluble SNARE was omitted instead of truncated (green and purple). This fusion with sQb3Δ and Qc3Δ occurs in stages, initially sensitive to antibody to either the HOPS SM subunit Vps33 or Sec18, then acquiring resistance to Vps33 antibody while remaining sensitive to the Sec18 ligand (*Figure 4—figure supplement 2*). When only the Qb-SNARE was membrane anchored, there was little fusion without Sec17 and Sec18 (*Figure 4E*, black curve). When the SNARE domain of sQa or Qc had been truncated, fusion was strictly dependent on non-hydrolyzable ATP analogs, and little fusion was seen with dual SNARE domain truncation. Similar patterns were seen with anchored Qc (*Figure 4*, I–L). We term the fusion induced by Sec17 and Sec18 in the presence of 3Δ SNARE domain truncations 'zippering bypass fusion'.

To explore the capacity of Sec17 and Sec18 to compensate for partial SNARE zippering, we assayed the fusion of Ypt7/R and Ypt7/Qa proteoliposomes with sQb3Δ, Qc3Δ, and HOPS, using various concentrations of wild-type or mutant Sec17, and with or without Sec18 and ATPγS (*Figure 5*). Without Sec17, fusion is not supported by Sec18 (A and B, blue curves). Limited fusion is possible with 1 or 2 μM wild-type Sec17 alone (A, black and red), but Sec18 allows faster fusion and with less Sec17 (A and B; tan). Fusion requires the apolar loop near the N-terminus of Sec17, as the

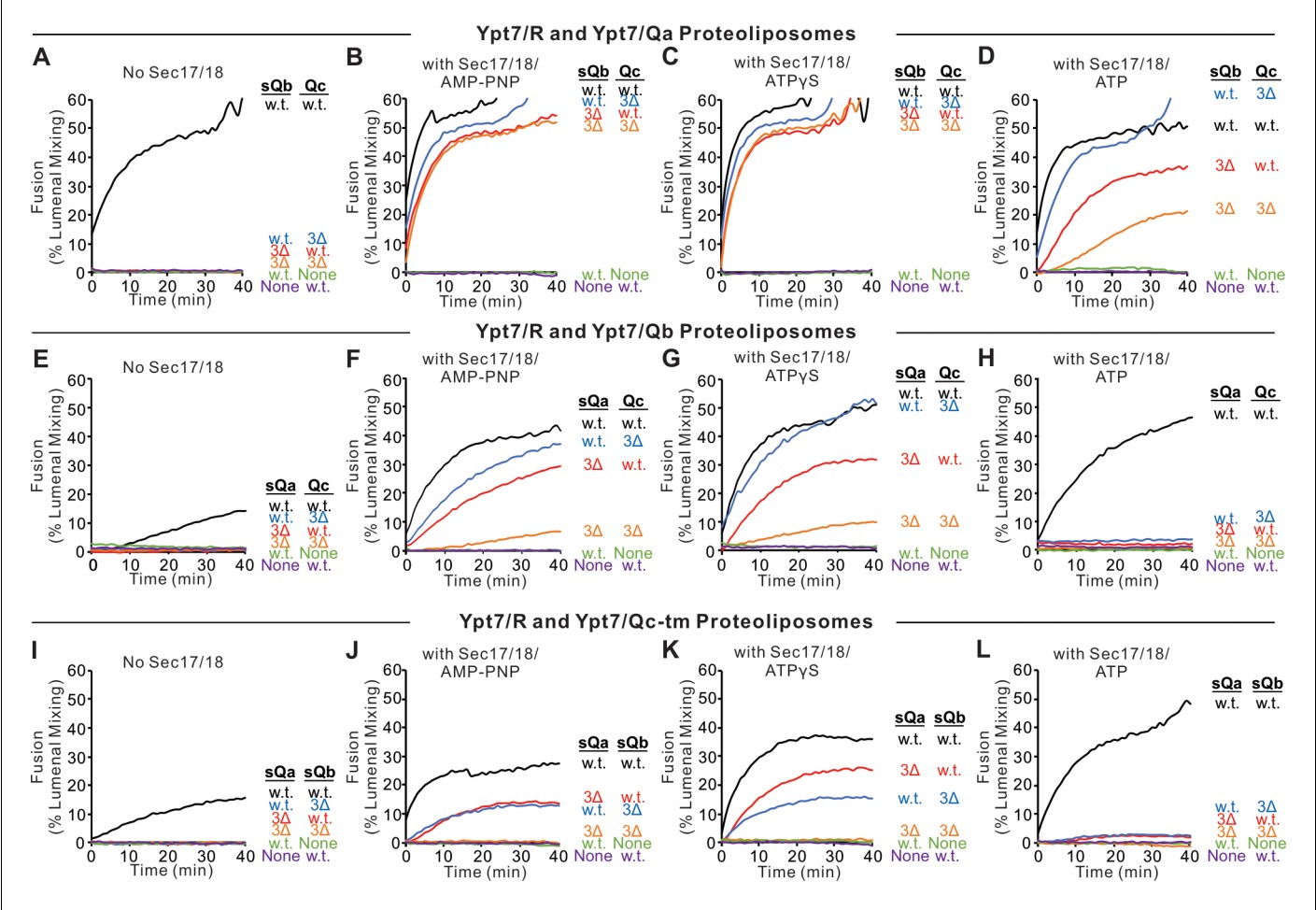

**Figure 4.** Fusion with single membrane-anchored Q-SNAREs. (**A–D**) Fusion incubations, as described in 'Materials and methods', had Ypt7/R and Ypt7/Qa proteoliposomes (1:8000 Ypt7-TM:lipid molar ratio, 1:16,000 SNARE:lipid molar ratio), 50 nM HOPS, 2 µM sQb or sQb3Δ, 2 µM Qc or Qc3Δ, and, where indicated, 600 nM Sec17, 300 nM Sec18, and 1 mM ATP, AMP-PNP, or ATPγS. (**E–H**) Fusion incubations as for (**A**), but with Ypt7/Qb proteoliposomes and 2 µM sQa or sQa3Δ, 2 µM Qc or Qc3Δ and Sec17, Sec18, and adenine nucleotide as indicated. (**I–L**) Fusion incubations as for (**A**), but with Ypt7/Qc-tm proteoliposomes and 2 µM sQa or sQa3Δ, 2 µM Qb or sQb3Δ and Sec17, Sec18, and adenine nucleotide as indicated. Mean and standard deviations from more than three independent experiments are shown in *Figure 4—figure supplement 1*.

The online version of this article includes the following source data and figure supplement(s) for figure 4:

**Source data 1.** Source data file (Excel) for *Figure 4A–L*.
**Figure supplement 1.** Statistics data for *Figure 4*.
**Figure supplement 2.** Selective inhibitors reveal successive fusion intermediates.

F21S,M22S mutation (FSMS) blocks fusion entirely (C and D). The K159E,K163E mutation (KEKE) diminishes Sec17:SNARE association (*Marz et al., 2003*). The KEKE mutation prevents fusion without Sec18 (E), but a slow and limited fusion with KEKE-Sec17 is restored by Sec18 (F). The C-terminal L291A,L292A mutation of Sec17 (LALA), which interferes with its cooperation with Sec18 for SNARE complex disassembly (*Barnard et al., 1997*; *Schwartz and Merz, 2009*; *Zick et al., 2015*), diminishes zippering bypass fusion (A vs. G, red curves), and a limited fusion is restored through the addition of Sec18 (H). These data suggest that Sec17 action directly requires its apolar loop domain, since the loss of this apolar region is not bypassed by Sec18. Sec18 may stimulate fusion by modulating the conformation of Sec17 associations with *trans*-SNARE complexes, but Sec18 is not simply promoting Sec17 binding, since it is still needed for fusion when Sec17 is joined to an integral N-terminal membrane anchor (*Figure 5—figure supplement 2*). Basic residues in the +3 to +8 layers near the C-terminus of the R and Qa SNARE domains are near acidic residues on the interior of the Sec17 assembly (*Figure 5I*). To determine whether Sec17 might rely on these ionic interactions to support

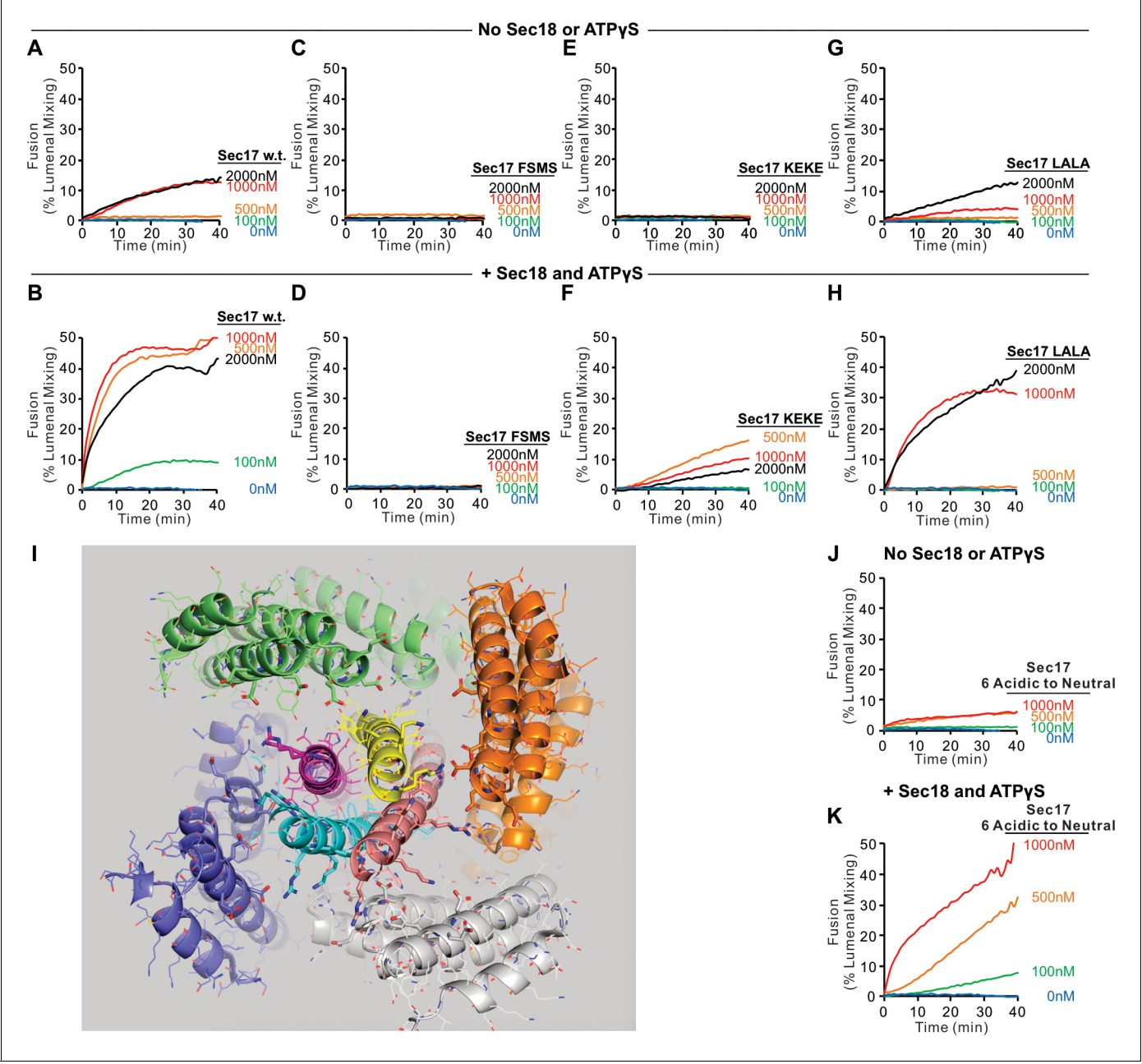

**Figure 5.** Role of each domain of Sec17 in zippering bypass fusion. (**A-H, J, K**) Fusion between Ypt7/R and Ypt7/Qa proteoliposomes (1:8000 Ypt7-TM: lipid and 1:16,000 SNARE/lipid molar ratios) was assayed with 50 nM HOPS, 2 μM sQb3Δ, 2 μM Qc3Δ, the indicated concentration of wild-type or mutant Sec17, and with or without 250 nM Sec18 and 1 mM ATPγS. (**I**) Ionic interactions between Sec17s and vacuolar SNAREs in the +4 to +8 layers in the model of the Sec18/Sec17/vacuolar SNARE complex (*Figure 1*). Colors: cyan: R; magenta: Qa; yellow: Qb; salmon: Qc; gray, orange, green, slate: Sec17; red: oxygen atoms; blue: nitrogen atoms. Cartoon representations are shown along with side chains shown as thin lines. Thick lines (sticks) are interacting glutamate and aspartate (acidic) residues on the surface of Sec17 (aminoacyl residues 34, 35, 38, 73, 74, 75) that interact with the vacuolar SNARE complex lysine and arginine basic residues in each of the four SNAREs. Mean and standard deviations from four independent experiments are shown in *Figure 5—figure supplement 1*.

The online version of this article includes the following source data and figure supplement(s) for figure 5:

**Source data 1.** Source data file (Excel) for *Figure 5A–H,J and K*.
**Figure supplement 1.** Mean and standard deviations of fusion after 60 min from four independent assays as in *Figure 5* are shown.
**Figure supplement 2.** Sec18 does not simply promote fusion by contributing to the affinity of Sec17 for membranes that bear SNARE complexes.

fusion, we converted these acidic residues of Sec17 to alanine or serine, creating the mutant Sec17-E34S,E35S,D38S,E73A,D74A,E75A (termed Sec17 6 Acidic to Neutral or Sec17-6AtoN), but this mutant Sec17 still supports fusion (*Figure 5*, J and K). Interestingly, mutation of acidic residues of αSNAP near the C-terminal end of the neuronal SNARE complex also only had a modest effect on disassembly activity of NSF/αSNAP (*Zhao et al., 2015*).

## Fusion despite triply-crippled SNARE zippering

Since SNARE zippering is driven by the sequestration of apolar residues into the interior of the 4-SNARE bundle, we examined the effect of converting the apolar residues of the Qa SNARE domain +4 to +8 layers to Ala, Ser, or Gly. Fusion between Ypt7/R and Ypt7/Qa proteoliposomes in the presence of HOPS, sQb, and Qc, but without Sec17 or Sec18 (*Figure 6A*, black curve), was diminished by replacing each of the +4 to +8 layer apolar residues of Qa with Ala (*Figure 6A*, green curve) and was abolished when they were replaced by Ser (red curve) or by Gly (blue curve). The persistence of some fusion with the Ala substitutions may reflect that two of the residues were already Ala, that Ala has the greatest propensity among the amino acids to form α-helices, Gly the least, and Ser is in-between (*Pace and Scholtz, 1998*), and that alanine itself is the least hydrophilic of these three amino acids. When these same incubations were performed with Sec17, Sec18, and ATPγS, rapid and comparable fusion was seen in each case (*Figure 6B*). When hydrolyzable ATP was

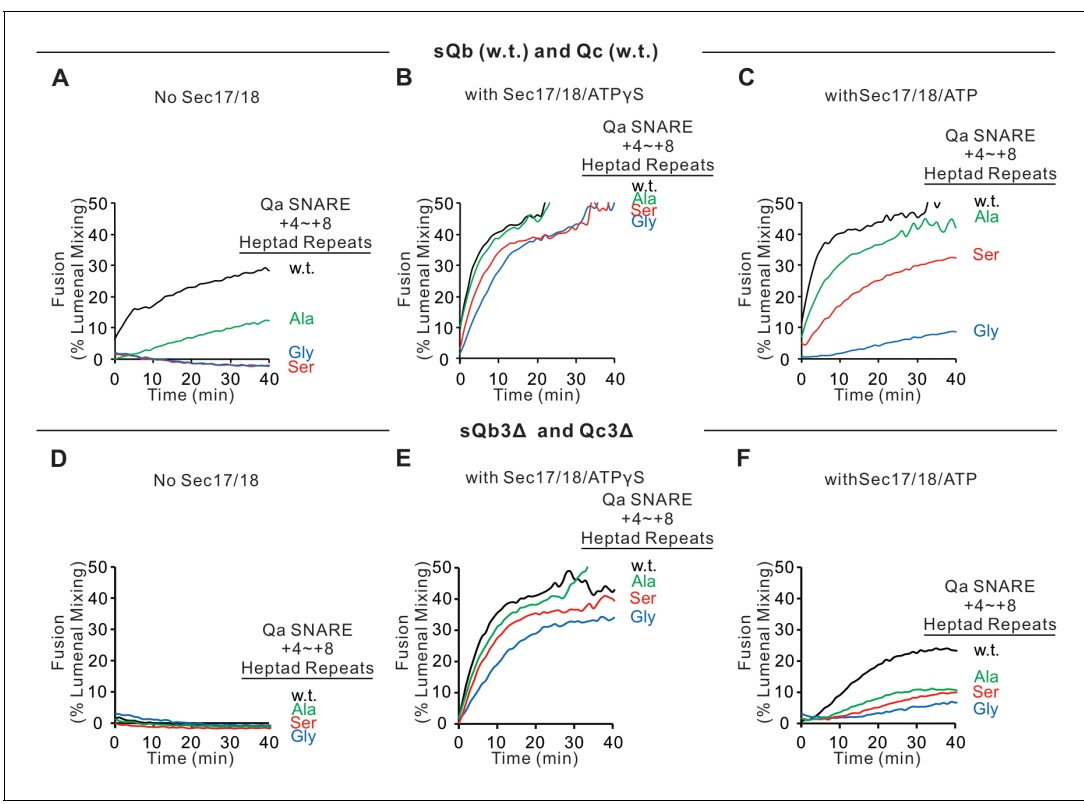

**Figure 6.** Sec17, Sec18, and ATPγS restore fusion to Ypt7/R and Ypt7/Qa proteoliposomes, which were triply-crippled from completion of SNARE domain zippering by deletion of the +4 to +8 layers of the sQb and Qc SNAREs and by substitution of the apolar residues of the +4 to +8 layers of Qa, substituting Ala, Ser, or Gly for L238, M242, A245, L249, and A252. Fusion incubations ('Materials and methods') had (**A–C**) Ypt7/R and Ypt7/Qa (w.t. (black), Gly mutant (blue), Ser mutant (red), or Ala mutant (green)) proteoliposomes (1:8000 Ypt7-TM:lipid molar ratio, 1:16,000 SNARE:lipid molar ratio), 50 nM HOPS, 2 μM sQb (w.t.), and 2 μM Qc (w.t.) (**A–C**) or (**D–F**) 2 μM Qc3Δ and 2 μM sQb3Δ. Sec17 and Sec18 buffers (**A and D**) or 600 nM Sec17, 300 nM Sec18, and 1 mM ATPγS (**B and E**) or ATP (**C and F**) were also present. Kinetics shown are representative of four experiments. Mean and standard deviations from four independent experiments are shown in *Figure 6—figure supplement 1*.

The online version of this article includes the following source data and figure supplement(s) for figure 6:

**Source data 1.** Source data file (Excel) for *Figure 6A–F*.

**Figure supplement 1.** Fusion assays between Ypt7/R- and Ypt7/Qa with Qa w.t. (black), Qa 5Ala mutant (green), Qa 5Gly mutant (blue), or Qa 5Ser mutant (red) as described in *Figure 6*.

used instead of ATPγS, there was little effect on the fusion kinetics with wild-type SNARE domain sequences (*Figure 6*, B vs. C, black curves). In contrast, hydrolyzable ATP led to fusion inhibition when SNARE packing stability was reduced by substitution of apolar residues by Ala, Ser, or Gly (*Figure 6C*). Though the apolar residues are not required for fusion aided by Sec17 and Sec18 (*Figure 6B*), they apparently stabilize the SNAREs against ATP-driven proofreading disassembly (*Figure 6C*). To triply weaken the completion of zippering, the same proteoliposomes with wild-type Qa or the Qa with +4 to +8 layers having small side-chain residues instead of apolar residues were incubated with HOPS, sQb3Δ, and Qc3Δ, either without Sec17 or Sec18 (*Figure 6D*) or with Sec17, Sec18, and ATPγS (*Figure 6E*) or ATP (*Figure 6F*). Fusion was optimally supported by Sec17, Sec18, and ATPγS (*Figure 6E*). The independence of this fusion from energy derived by zippering is underscored by the similar fusion rates in all incubations with Sec17, Sec18, and ATPγS, whether with apolar or polar Qa +4 to +8 residues or with full-length or +4 to +8-truncated sQb and Qc (*Figure 6*, B vs. E). With the Qb and Qc SNARE domains truncated, and the Qa lacking apolar inward-facing amino acyl side chains, little or no energy could be gained from the completion of R and mutant-Qa zippering. Thus, Sec17 acts in three ways: triggering a zippering-dependent SNARE conformational change in the presence of HOPS and full-length SNARE domains (*Figure 2*), acting with Sec18 to promote fusion independent of energy from zippering (*Figure 6*), and supporting the disassembly of post-fusion *cis*-SNARE complexes by Sec18.

## Discussion

The catalytic roles of fusion proteins have been gleaned from functional reconstitution studies. These studies initially showed that the zippering of concentrated SNAREs can drive slow fusion (*Weber et al., 1998*; *Fukuda et al., 2000*). As proteoliposome SNARE levels were lowered toward physiological levels, reconstituted vacuolar and neuronal fusion reactions required additional factors (*Zick and Wickner, 2016*; *Stepien and Rizo, 2021*). In addition to SNAREs, reconstituted neuronal fusion requires Munc18, Munc13, calcium, NSF, and αSNAP (*Ma et al., 2013*; *Lai et al., 2017*) while reconstituted vacuole fusion needs HOPS (*Stroupe et al., 2006*), the Rab Ypt7 (*Stroupe et al., 2009*; *Ohya et al., 2009*), and specific lipid head-group composition and fatty acyl fluidity (*Zick and Wickner, 2016*). HOPS and other SM proteins catalyze SNARE assembly (*Baker et al., 2015*; *Orr et al., 2017*; *Jiao et al., 2018*), regulated by an activated Rab (*Zick and Wickner, 2016*; *Torng et al., 2020*), and may confer resistance to Sec17/αSNAP- and Sec18/NSF-mediated *trans*-SNARE disassembly (*Xu et al., 2010*; *Jun and Wickner, 2019*). Sec17/αSNAP and Sec18/NSF stimulate fusion with HOPS and wild-type SNAREs (*Mima et al., 2008*; *Song et al., 2017*).

Fusion can be supported by either complete 4-SNARE zippering without Sec17 or Sec18, or by Sec17 and Sec18 association with only partially zippered SNAREs, but the most rapid fusion requires both (*Song et al., 2017*). While Sec17 and Sec18 are known to bypass the fusion blockade by Qc C-terminal truncation alone (*Schwartz and Merz, 2009*; *Schwartz et al., 2017*), the generality of this bypass with respect to any one Q-SNARE or even two Q-SNAREs (*Figures 3* and *4*) shows that it is not specific to Qc alone. Moreover, when zippering with full-length SNARE domains is weakened by the substitution of small amino acyl residues for large apolar residues in the +4 to +8 layers of Qa, Sec17 and Sec18 will also restore fusion (*Figure 6A,B*). Strikingly, Sec17 and Sec18 drive fusion despite a triple blockade to complete zippering, namely the absence of two C-terminal Q-SNARE domains and the lack of apolarity in the third (*Figure 6E*). In the NSF/αSNAP/SNARE complex, the 4-SNAREs wrap around each other in a left-handed super helix, and the Sec17s wrap around them in a right-handed fashion, yet they form a specific structure (*Zhao et al., 2015*; *White et al., 2018*). It seems unlikely that residues from one or more Sec17 could substitute for the missing residues when two heptads are removed from the C-termini of one or even two Q-SNAREs and the bulky apolar residues are removed from the third.

From the earliest reconstitutions of HOPS-dependent fusion (*Mima et al., 2008*) and in subsequent studies (e.g., *Zick et al., 2015*), Sec17 and Sec18 gave strong stimulation. In contrast, Sec17 and Sec18 inhibit SNARE-only fusion, or fusion with non-physiological tethers (*Mima et al., 2008*; *Zick and Wickner, 2014*; *Schwartz et al., 2017*; *Song and Wickner, 2019*). HOPS not only binds each SNARE, but also has direct affinity for Sec17 (*Figure 2—figure supplement 2*). In model studies with *cis*-SNARE complexes, HOPS is necessary for Sec17 to enhance zippering per se

(*Figure 2C*). HOPS:Sec17 binding may underlie their synergy for fusion, but further studies are needed to test this idea.

Earlier studies and our current work suggest a working model of vacuole membrane fusion, encompassing findings here and elsewhere (*Figure 7*). HOPS exploits the affinity of its Vps39 and Vps41 subunits for the Rab Ypt7 (*Brett et al., 2008*) on each fusion partner membrane (*Figure 7A*) to mediate (Step 1) tethering (*Hickey and Wickner, 2010*). Tethered membranes (*Figure 7B*) are a prerequisite for SNARE assembly in an active conformation, likely a common N to C SNARE domain orientation (*Song and Wickner, 2019*). HOPS has direct affinity for each of the four vacuolar SNAREs (*Song et al., 2020*) and is allosterically activated by vacuolar lipids and Ypt7:GTP (*Torng et al., 2020*) as a catalyst of SNARE assembly (Step 2). SNAREs begin to zipper (*Figure 7C*) from the N- toward the C-terminal end of their SNARE domain. When any one Q-SNARE is omitted, fusion intermediates assemble, which undergo very rapid fusion when the missing Q-SNARE is supplied (*Song et al., 2020*), suggesting that fusion without Sec17/Sec18 is rate-limited by the

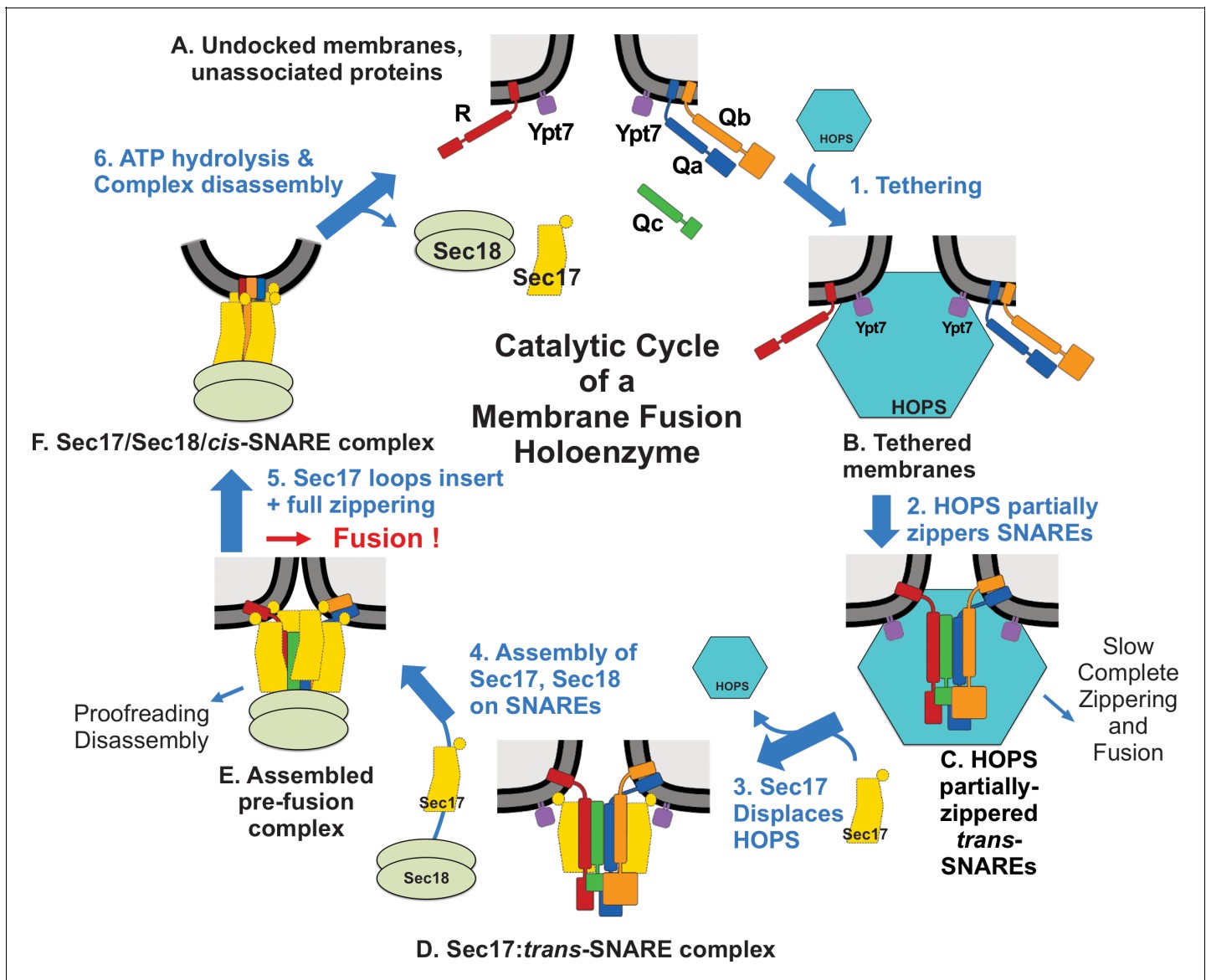

**Figure 7.** Current working model. Catalyzed tethering and SNARE assembly (**A–C**) is followed by HOPS displacement by Sec17 (**C and D**) and further assembly of Sec17 and Sec18 on the partially zippered SNARE complex, promoting completion of zippering and apolar Sec17 loop insertion (**D–F**), both of which promote fusion. See text for discussion.

completion of SNARE zippering and/or the spontaneous release of bulky HOPS. Sec17 has direct affinity for SNAREs (*Söllner et al., 1993*), HOPS (*Figure 2—figure supplement 2*), lipids (*Clary et al., 1990*; *Zick et al., 2015*), and Sec18 (*Söllner et al., 1993*). Sec17 displaces HOPS from the SNAREs (*Figure 7*, Step 3), as shown by earlier studies: (1) vacuolar SNAREs are found in complex with Sec17 or HOPS, but not with both, and Sec17 can displace HOPS from SNAREs (*Collins et al., 2005*); (2) when vacuolar HOPS-dependent reconstituted fusion is arrested by Qc3Δ, HOPS is bound to the incompletely zippered SNARE complex until it is displaced by Sec17 (*Schwartz et al., 2017*); and (3) *trans*-SNARE complexes, which assemble between isolated vacuoles, are largely associated with Sec17 (*Xu et al., 2010*). The Sec17 association with partially zippered *trans*-SNARE complex (*Figure 7D*) promotes a conformational change (*Figure 2*) leading to complete zippering. The Sec7: 4-SNARE complex will bind Sec18 (Step 4) through its affinities for both SNAREs (*Zick et al., 2015*) and for Sec17. Sec18 regulates, in some unknown fashion, the Sec17/α SNAP assembly into an oligomeric structure surrounding the SNARE complex intermediate (*Zhao et al., 2015*), a *trans*-anchored Sec18/Sec17/SNARE pre-fusion complex (*Figure 7E*). Some ATP hydrolysis-dependent disassembly can occur, which may represent proofreading of incorrect and unstable *trans*-SNARE complexes (*Choi et al., 2018*). Sec17 oligomerization may also be stabilized or guided by the insertion of its N-terminal loop into the membranes. While SNAREs can slowly complete zippering and support fusion without Sec17 or Sec18 (*Mima et al., 2008*; *Song et al., 2017*), and slow fusion can occur with Sec17 and Sec18 when sQb and Qc lack the C-terminal portion of their zippering domain (*Figure 4D*), optimal fusion requires four complete SNARE domains, Sec17/αSNAP, and Sec18/NSF. The energy for fusion (Step 5) derives from multiple sources: the completion of SNARE zippering, the binding energies which create the Sec17 structure surrounding the SNAREs, and the energy of bilayer distortion through Sec17 apolar loop insertion. *Cis*-SNARE complexes (*Figure 7F*) formed by fusion (*Söllner et al., 1993*; *Mayer et al., 1996*) are disassembled by Sec17, Sec18, and ATP (Step 6), freeing SNAREs from each other (*Figure 7A*) for later assembly in *trans*. Each of these components – the four SNAREs, the Rab Ypt7, HOPS, Sec17, and Sec18 – is part of this ordered pathway of associations and disassembly, constituting a holoenzyme for vacuole membrane fusion.

Our current studies reveal a general capacity of Sec17/αSNAP and Sec18/NSF to support fusion, even when little or no energy would be derived from completion of zippering. Sec17 provides a cage-like environment (*Chakraborty et al., 2010*), albeit with side fenestrations (*Schwartz et al., 2017*), which may favor SNARE zippering or, where zippering is blocked, allow the remaining SNARE domains to attain positions and conformations that approximate zippering. The apolar N-domains of the four Sec17s are clearly essential for fusion (*Figure 5* and *Figure 5—figure supplement 2*), whether through positioning the Sec17s, facilitating their assembly, or directly inserting as a 'membrane wedge' and thereby contributing to bilayer disruption at the fusion site. The continued need for this apolar loop when Sec17 is integrally membrane-anchored (*Figure 5—figure supplement 2*) shows that it does not simply contribute to Sec17 membrane association. Elements of 'Sec17 cage' and 'membrane wedge' action are not mutually exclusive. Further work will be needed to determine the relative energies of Sec17 and Sec18 binding to the assembling 4-SNARE *trans*-complex, each of the four Sec17s inserting its apolar loops into the membrane, Sec17 forming lateral associations with other Sec17 molecules in the cage around the SNAREs, and complete SNARE zippering, as each of these helps to achieve the bilayer distortions of fusion.

Our current studies further our understanding of how Sec17(αSNAP), Sec18 (NSF), and SNAREs cooperate to promote membrane fusion. Sec17 has multiple functions: (a) It supports Sec18 association with SNARE complexes for their ATP-driven disassembly for subsequent rounds of fusion (*Ungermann et al., 1998*; *Choi et al., 2018*). Sec17/αSNAP and Sec18/NSF are also part of a quality control system that, for vacuoles, includes HOPS (*Starai et al., 2008*) and, in the neuronal system, includes Munc18 and Munc13 (*Ma et al., 2013*; *Lai et al., 2017*). These disassembly reactions require ATP hydrolysis and can be interrupted by the Sec17 LALA mutation. (b) Sec17 promotes a conformational change that leads to complete SNARE zippering. While neuronal SNAREs, which are properly assembled (i.e., involving the Munc18/Munc13 pathway), will fully zipper with a high energy yield, this may not be true for all non-neuronal SNAREs. (c) Sec17 promotes fusion even when SNARE zippering is incomplete. Sec17 uses the partially zippered SNAREs as a platform to bind to the fusion site and its apolar N-domain loop to trigger fusion. Sec18 also has multiple functions: (a) Sec18/NSF drives ATP-driven SNARE disassembly (*Zhao et al., 2015*), blocked by the Sec17 LALA

mutation. (b) Sec18 supports Sec17 for direct promotion of fusion, built on a platform of partially zippered SNAREs. Note that the loading of SNARE N-terminal residues into the pore of the D1 ring of NSF does not require hydrolysis (*White et al., 2018*), so initial assembly of the NSF/α SNAP/SNARE complex occurs in the absence of hydrolysis. This does not need ATP hydrolysis and is insensitive to the LALA mutation. Thirdly, SNAREs also have multiple functions: (a) They can completely zipper, thereby stressing the bilayer and triggering fusion (*Weber et al., 1998*; *Sutton et al., 1998*). As previously reported (*Mima et al., 2008*; *Song et al., 2017*), HOPS alone, without Sec17 or Sec18, will support slow SNARE-dependent fusion. (b) SNAREs support the assembly of a microdomain in which multiple fusion proteins and lipids become highly enriched (*Fratti et al., 2004*). (c) As shown here, SNAREs form an essential platform for Sec17 and Sec18 to contribute to fusion independently of completion of zippering. With physiological levels of wild-type vacuolar SNAREs, rapid fusion requires Rab-activated HOPS, Sec17, Sec18, and ATP.

While intracellular fusion reactions share many requirements, such as for SNAREs and SM protein, there are major differences as well. Synaptic vesicle fusion and other calcium-dependent secretion systems require calcium, synaptotagmin, Munc13, and complexin, none of which have their obvious counterparts in calcium-independent systems. Vacuole and endosome fusion have their SM protein as an integral subunit of the tethering complex, which is not seen in other organelles. The similarities and differences in fusion pathways at each organelle will be clarified as each is more thoroughly studied.

# Materials and methods

**Key resources table**

| Reagent type (species) or resource | Designation | Source or reference | Identifiers | Additional information |
|---|---|---|---|---|
| Gene (*Saccharomyces cerevisiae*) | Nyv1 | *Saccharomyces* Genome Database | SGD:S000004083 | |
| Gene (*Saccharomyces cerevisiae*) | Vam3 | *Saccharomyces* Genome Database | SGD:S000005632 | |
| Gene (*Saccharomyces cerevisiae*) | Vti1 | *Saccharomyces* Genome Database | SGD:S000004810 | |
| Gene (*Saccharomyces cerevisiae*) | Vam7 | *Saccharomyces* Genome Database | SGD:S000003180 | |
| Gene (*Saccharomyces cerevisiae*) | Ypt7 | *Saccharomyces* Genome Database | SGD:S000004460 | |
| Gene (*Saccharomyces cerevisiae*) | Sec17 | *Saccharomyces* Genome Database | SGD:S000000146 | |
| Gene (*Saccharomyces cerevisiae*) | Sec18 | *Saccharomyces* Genome Database | SGD:S000000284 | |
| Gene (*Saccharomyces cerevisiae*) | Vps33 | *Saccharomyces* Genome Database | SGD:S000004388 | |
| Gene (*Saccharomyces cerevisiae*) | Vps39 | *Saccharomyces* Genome Database | SGD:S000002235 | |
| Gene (*Saccharomyces cerevisiae*) | Vps41 | *Saccharomyces* Genome Database | SGD:S000002487 | |

*Continued on next page*

*Continued*

| Reagent type (species) or resource | Designation | Source or reference | Identifiers | Additional information |
|---|---|---|---|---|
| Gene (*Saccharomyces cerevisiae*) | Vps16 | *Saccharomyces* Genome Database | SGD:S000005966 | |
| Gene (*Saccharomyces cerevisiae*) | Vps11 | *Saccharomyces* Genome Database | SGD:S000004844 | |
| Gene (*Saccharomyces cerevisiae*) | Vps18 | *Saccharomyces* Genome Database | SGD:S000004138 | |
| Peptide, recombinant protein | GST-R (Nyv1) | PMID:18650938 | | Purified from *E. coli*. |
| Peptide, recombinant protein | GST-Qa (Vam3) | PMID:18650938 | | Purified from *E. coli*. |
| Peptide, recombinant protein | GST-Qb (Vti1) | PMID:18650938 | | Purified from *E. coli*. |
| Peptide, recombinant protein | His-R (Nyv1) | PMID:22174414 | | Purified from *E. coli*. |
| Peptide, recombinant protein | GST-sQa (soluble) | PMID:28637767 | | Purified from *E. coli*. |
| Peptide, recombinant protein | MBP-sQb (soluble) | PMID:24088569 | | Purified from *E. coli*. |
| Peptide, recombinant protein | GST-Qb (Vti1)3Δ | This study | | Purified from *E. coli*. |
| Peptide, recombinant protein | GST-Qa (Vam3)3Δ | This study | | Purified from *E. coli*. |
| Peptide, recombinant protein | Qc (Vam7) | PMID:17699614 | | Purified from *E. coli*. |
| Peptide, recombinant protein | Qc (Vam7)-tm | PMID:23071309 | | Purified from *E. coli*. |
| Peptide, recombinant protein | Qc (Vam7) C208S, M250C | This study | | Purified from *E. coli*. |
| Peptide, recombinant protein | Qc (Vam7) C208S, A316C | This study | | Purified from *E. coli*. |
| Peptide, recombinant protein | MBP-Cys-sQb (SNARE domain) | PMID:28637767 | | Purified from *E. coli*. |
| Peptide, recombinant protein | MBP-Cys-sQb (SNARE domain) | PMID:28637767 | | Purified from *E. coli*. |
| Peptide, recombinant protein | Ypt7 | PMID:24088569 | | Purified from *E. coli*. |

*Continued on next page*

*Continued*

| Reagent type (species) or resource | Designation | Source or reference | Identifiers | Additional information |
|---|---|---|---|---|
| Peptide, recombinant protein | Ypt7-TM | PMID:31235584 | | Purified from *E. coli*. |
| Peptide, recombinant protein | Sec17 | PMID:19414611 | | Purified from *E. coli*. |
| Peptide, recombinant protein | Sec17 FSMS | PMID:28925353 | | Purified from *E. coli*. |
| Peptide, recombinant protein | Sec17 KEKE | PMID:28925353 | | Purified from *E. coli*. |
| Peptide, recombinant protein | Sec17 LALA | PMID:19414611 | | Purified from *E. coli*. |
| Peptide, recombinant protein | GST-Sec17 6AtoN | This study | | Purified from *E. coli*. |
| Peptide, recombinant protein | GST-Sec17-TM | PMID:28718762 | | Purified from *E. coli*. |
| Peptide, recombinant protein | GST-Sec17-TM FSMS | PMID:28718762 | | Purified from *E. coli*. |
| Peptide, recombinant protein | TEV protease | PMID:18007597 | | Purified from *E. coli*. |
| Peptide, recombinant protein | HOPS | PMID:18385512 | | Purified from *Saccharomyces cerevisiae*. |
| Antibody | Anti-Vam3 (rabbit polyclonal) | PMID:12566429 | Wickner lab stock | WB: 1:2000 |
| Antibody | Anti-Nyv1 (rabbit polyclonal) | PMID:10385523 | Wickner lab stock | WB: 0.65 µg/ml |
| Antibody | Anti-Vti1 (rabbit polyclonal) | PMID:18007597 | Wickner lab stock | WB: 0.47 µg/ml |
| Antibody | Anti-Vam7 (rabbit polyclonal) | PMID:14734531 | Wickner lab stock | WB: 0.1 µg/ml |
| Antibody | Anti-Vps16 (rabbit polyclonal) | PMID:18007597 | Wickner lab stock | WB: 1 µg/ml |
| Antibody | Anti-Vps33 (rabbit polyclonal) | PMID:10944212 | Wickner lab stock | Inhibition assay: 1 µg |
| Antibody | Anti-Sec18 (rabbit polyclonal) | PMID:11483507 | Wickner lab stock | Inhibition assay: 1 µg |
| Chemical compound, drug | Cy5-derivatized streptavidin | SeraCare Life Sciences | 5270–0023 | |
| Chemical compound, drug | Biotinylated PhycoE | Thermo Fisher Scientific | p811 | |
| Chemical compound, drug | Streptavidin | Thermo Fisher Scientific | 434302 | |
| Chemical compound, drug | 1,2-dilinoleoyl-sn-glycero-3-phosphocholine | Avanti polar lipids | 850385 | |
| Chemical compound, drug | 1,2-dilinoleoyl-sn-glycero-3-phospho-L-serine | Avanti polar lipids | 840040 | |
| Chemical compound, drug | 1,2-dilinoleoyl-sn-glycero-3-phosphoethanolamine | Avanti polar lipids | 850755 | |
| Chemical compound, drug | 1,2-dilinoleoyl-sn-glycero-3-phosphate | Avanti polar lipids | 840885 | |
| Chemical compound, drug | L-α-phosphatidylinositol | Avanti polar lipids | 840044 | |

*Continued on next page*

*Continued*

| Reagent type (species) or resource | Designation | Source or reference | Identifiers | Additional information |
|---|---|---|---|---|
| Chemical compound, drug | 1,2-dipalmitoyl-sn-glycerol | Avanti polar lipids | 800816 | |
| Chemical compound, drug | Ergosterol | Sigma | 45480 | |
| Chemical compound, drug | PI(3)P | Echelon Bioscience | P-3016 | |
| Chemical compound, drug | Rhodamine DHPE | Invitrogen | L1392 | |
| Chemical compound, drug | NBD-PE | Invitrogen | N360 | |
| Chemical compound,drug | Marina-blue | Invitrogen | M12652 | |
| Software and algorithms | UN-SCAN-IT | Silk Scientific | | |
| Chemical compound, drug | Alexa Fluor 568 C5-maleimide | Thermo Fisher Scientific | A20341 | |
| Chemical compound, drug | Oregon Green 488 Maleimide | Thermo Fisher Scientific | O6034 | |
| Chemical compound, drug | Oregon Green 488 Maleimide | Thermo Fisher Scientific | O6034 | |
| Chemical compound, drug | Pierce TCEP-HCl | Thermo Fisher Scientific | 20490 | |
| Chemical compound, drug | L-cysteine | Sigma-Aldrich | 30089 | |

PI3P was from Echelon (Salt Lake City, Utah), ergosterol from Sigma (St. Louis, MO), fluorescent lipids from Thermo Fisher (Waltham, MA), and other lipids were from Avanti (Alabaster, AL). Biobeads SM2 were from BioRad, Cy5-Streptavidin from SeraCare (Milford, MA), biotinylated phycoerythrin from Invitrogen (Eugene, OR), and underivatized streptavidin from Thermo Fisher. Spectrapor six dialysis tubing (7.5 mm diameter, 25 kDa cutoff) was from Spectrum Labs (Las Vegas, NV). Octyl-b-D-glucopyranoside was purchased from Anatrace (Maumee, OH).

## Mutant constructions

Sec17 with six acidic amino acids mutated to neutral residues, GST-Sec17 (E34S, E35S, D38S, E73A, D74A, E75A), was generated by PCR with Phusion high-fidelity DNA polymerase (NEB). The DNA fragment was cloned into BamHI- and SalI-digested pGST parallel1 vector (*Sheffield et al., 1999*) with an In-Fusion kit (Takara Bio USA, Mountain View, CA). Using inverse PCR, pParallel1-GST-Sec17 mutant (E34S, E35S, and D38S) was amplified with Phusion high-fidelity DNA polymerase (NEB) from a GST-Sec17 construct. The amplified linear DNA was re-circularized with an In-fusion kit (Takara Bio USA). To generate Sec17 acidic to neutral mutants, pParallel1-GST-Sec17 mutant (E34S, E35S, and D38S) was amplified with a Sec17 E73A, D74A, and E75A mutant primer set using Phusion high-fidelity DNA polymerase (NEB) and re-circularized with an In-fusion kit (Takara Bio USA).

> For Sec17-E34S,E35S, D38S,
> F: TCGTCGGCTGCTTCTCTTTGTGTCCAAGCAGCCAC
> R: AGAAGCAGCCGACGAAAACTTGTATGAATCAGAAC
> For Sec17 E73A, D74A, E75A,
> F: GGTAATGCAGCCGCAGCAGGAAATACCTACGTAGA
> R: TGCGGCTGCATTACCAGCCTTTTTCTGATAGTCAG

GST-sQa3Δ with amino acyl residues 1–235 and GST-sQb3Δ with amino acyl residues 1–160 were generated by PCR with Phusion high-fidelity DNA polymerase (NEB). DNA fragments were cloned into BamHI- and SalI-digested pGST parallel1 vector (*Sheffield et al., 1999*) with an In-Fusion kit (Takara Bio USA).

For GST-sQa3Δ,
F: AGGGCGCCATGGATCCGATGTCCTTTTTCGACATCGA
R: AGTTGAGCTCGTCGACTAGATATTCTCGTCTATGGTGG
For GST-sQb3Δ,
F: AGGGCGCCATGGATCCGATGAGTTCCCTATTAATA
R: AGTTGAGCTCGTCGACTACAAGGTCTGTCTTGCATTTT

To generate Qa with L238, M242, A245, L249, and A252 changed to Ala, Ser, or Gly, the pParallel1 GST vector with Qa lacking residues 228–257 was generated by inverse PCR with pParallel1 GST-Qa (*Mima et al., 2008*) using Phusion high-fidelity DNA polymerase (NEB). The DNA duplex with mutations (Gly, Ser, or Ala) of the +4 to +8 heptad repeats was cloned into the amplified vector bearing Qa 1–227 with an In-fusion kit (Takara Bio USA, Mountain View, CA).

## For inverse PCR of pParallel1 GST and Qa 1–227,

F: GACCAGCATCAGAGGGACCG
R: GTCTATGGTGGTTACTTGTT

## For the Gly mutant of Qa,

Sequence 1: GTAACCACCATAGACGAGAATATCTCGCATGGCCATGATAACGGCCAGAA
TGGCAACAAACAAGGCACCAGAGGCGACCAGCATCAGAGG
Sequence 2: CCTCTGATGCTGGTCGCCTCTGGTGCCTTGTTTGTTGCCATTCTGGCCGTTA
TCATGGCCATGCGAGATATTCTCGTCTATGGTGGTTAC

## For the Ser mutant of Qa,

Sequence 1: GTAACCACCATAGACGAGAATATCTCGCATAGCCATGATAACAGCCAGAA
TAGCAACAAACAAAGCACCAGAAGCGACCAGCATCAGAGG
Sequence 2: CCTCTGATGCTGGTCGCTTCTGGTGCTTTGTTTGTTGCTATTCTGGCTGTTATCA
TGGCTATGCGAGATATTCTCGTCTATGGTGGTTAC

## For the Ala mutant of Qa,

Sequence 1: GTAACCACCATAGACGAGAATATCTCGCATGCCCATGATAACGCCCAGAA
TGCCAACAAACAAGCCACCAGAGCCGACCAGCATCAGAGG
Sequence 2: CCTCTGATGCTGGTCGGCTCTGGTGGCTTGTTTGTTGGCATTCTGGGCGTTA
TCATGGGCATGCGAGATATTCTCGTCTATGGTGGTTAC

Vam7 mutants with cysteines inserted at the N- and C-termini of the SNARE domain were generated by inverse PCR with the Vam7 intein vector (*Schwartz and Merz, 2009*) and Phusion high-fidelity DNA polymerase (NEB). First, the native cysteine was removed by mutating it to serine (C208S) using the mutant primer set below. Vam7 mutants with cysteines inserted near the N- and C-termini of the SNARE domain, M250C and A316C, respectively, were generated from the cysteine-lacking plasmid in the same fashion.

For Vam7-C208S,
F: GAAAGCGATGACATTGGTACAGCAAACATAGCTCA
R: CAATGTCATCGCTTTCCTTGAGCAAGGACCTCAAT
For Vam7-C208S,M250C (Qc with N-terminal cysteine),
F: GGGCAGTGTCAAATGGTGCGCGATCAAGAACAA
R: CCATTTGACACTGCCCCTGTTGCAAATCGTTAT
For Vam7-C208S,A316C (Qc with C-terminal cysteine),
F: CAACAGTTGTTGAATTCTCGAGCACCACCA
R: CAACAACTGTTGTTAAAATGTCTAGCCTTCTTGTTGGC

## Protein isolation

HOPS and prenyl-Ypt7 (*Zick and Wickner, 2013*), Ypt7-TM (*Song et al., 2020*), Sec17 (*Schwartz and Merz, 2009*), TM-anchored Sec17 and TM-anchored Sec17-F21SM22S (*Song et al., 2017*), Sec18 (*Mayer et al., 1996*), wild-type vacuolar SNAREs and his6-R (*Mima et al., 2008*; *Schwartz and Merz, 2009*; *Zucchi and Zick, 2011*; *Izawa et al., 2012*), sQb (*Zick and Wickner, 2013*) Qc-C208SM250C, Qc-C208SA316C, and Qc3Δ (*Schwartz and Merz, 2009*) were purified as described. sQa, sQa3Δ, and sQb3Δ were purified by a modification of prior methods (*Zick and Wickner, 2013*; *Song et al., 2020*). pGST-Parallel1 with sQa, sQa3Δ, or sQb3Δ was transformed into Rosetta2 (DE3)-pLysS cells (EMD Millipore, Billerica, MA). Luria–Bertani (LB) broth (100 ml) containing 100 µg/ml ampicillin and 34 µg/ml chloramphenicol was inoculated with a single colony. After overnight incubation with shaking at 37°C, 40 ml portions of the preculture were added to two 6 l flasks, each with 3 l of LB medium, containing 100 µg/ml ampicillin and 34 µg/ml chloramphenicol and shaken (200 rpm) at 37°C to an OD600 of 0.8. Isopropyl β-D-1-thiogalactopyranoside was added to 0.5 mM. After 3 hr of continued shaking at 37°C, bacteria were harvested by centrifugation (5000 rpm, 5 min, 4°C). Cell pellets were resuspended in 60 ml of 20 mM HEPES-NaOH (pH 7.4), 200 mM NaCl, 1 mM EDTA, 1 mM dithiothreitol (DTT), 200 mM phenylmethyl sulfonylfluoride, and 1 X protease inhibitor cocktail (*Xu and Wickner, 1996*). Resuspended cells were passed twice through a French press at 900 psi. The cell lysate was centrifuged (4°C, 1 hr, 50,000 rpm, SW 60Ti rotor [Beckman Coulter, Brea, CA]). The supernatant was added to 20 ml of Glutathione Agarose 4B resin (Genesee Scientific, San Diego, CA), which had been equilibrated with wash buffer (20 mM HEPES-NaOH (pH 7.4), 200 mM NaCl, and 1 mM EDTA, 1 mM DTT) and nutated at 4°C for 2 hr. The suspended resin was poured into a 2.5-cm-diameter column, drained, and washed with 100 ml wash buffer. The GST-tagged protein was eluted with 20 mM HEPES-NaOH (pH 7.4), 200 mM NaCl, 1 mM EDTA, 1 mM DTT, 5% glycerol, and 20 mM glutathione, and the GST tag removed by TEV protease.

## Proteoliposome fusion

Proteoliposomes were prepared by detergent dialysis from β-octylglucoside-mixed micelles for fusion assays (*Song et al., 2017*; *Song and Wickner, 2019*) and SNARE assembly assays (*Torng et al., 2020*) as described. Briefly, for the fusion assay, proteoliposomes (1 mM lipid) were prepared with membrane-bound Ypt7 and R at 1:8000 and 1:16,000 molar ratios to lipid, respectively, and with lumenal biotin-phycoerythrin. Proteoliposomes were also prepared with membrane-bound Ypt7 and the indicated Q-SNAREs at 1:8000 and 1:16,000 molar ratios to lipid, respectively, plus lumenal Cy5-streptavidin. These were incubated separately for 10 min at 27°C with 1 µM GTP and 1 mM EDTA, and then MgCl$_2$ was added to 2 mM. After prewarming (27°C for 10 min) the separately GTP-exchanged proteoliposomes in a 384-well plate, fusion reactions were initiated by mixing 5 µl of each proteoliposome preparation and supplementing with other fusion factors in volumes summing to 10 µl, continuing incubation at 27°C in a Spectramax fluorescent plate reader. Fusion incubations (20 µl) in RB150 (20 mM HEPES/NaOH, pH 7.4, 150 mM NaCl, 10% glycerol) had proteoliposomes (0.5 mM total lipid concentration), 50 nM HOPS, the indicated concentrations of sSNAREs, 400 or 600 nM Sec17, 300 nM Sec18, 1 mM ATP or its analogs, and 3 mM MgCl$_2$, as modified in each figure legend.

## SNARE assembly assay

Assays were performed as described previously (*Torng et al., 2020*) with one addendum. In brief, reactions (20 µl) were performed at 27°C in a SpectraMax Gemini XPS (Molecular Devices) plate reader. Standard reactions include HOPS (160 nM), proteoliposomes (0.5 mM lipid, with SNARE and Ypt7 at a molar ratio of either 1:2000 or 1:4000 for Ypt7/R proteoliposomes and Ypt7/RQa proteoliposomes, respectively), and fluorescently labeled Qb and Qc (1 µM), and sQa (1 µM) as necessary. These were incubated for 60 min, and then Sec17 was added to 500 nM. Three fluorescence channels were read simultaneously at intervals of 30 s: the donor channel Oregon Green 488 from Qc (excitation [ex]: 497 nm; emission [em]: 527 nm; cutoff [c/o]: 515 nm), the acceptor channel Alexa Fluor 568 from Qb (ex: 568 nm; em: 605 nm; c/o: 590 nm), and the FRET channel (ex: 490 nm; em: 615 nm; c/o: 590 nm). For each time point, the bleed through-corrected FRET signal was obtained by subtracting the background signals coming from the donor and acceptor channels from the signal in the FRET channel as detailed in *Torng et al., 2020*. This was further corrected by

dividing by the geometric mean of the donor and acceptor signals. The final corrected signal, reported as 'Average FRET efficiency,' is a combined measure of the proportion of fluorescent SNAREs undergoing FRET and their average FRET efficiency.

## Molecular models

MODELLER (*Webb and Sali, 2016*) was used to create individual homology models of the vacuolar SNARE complex (Nyv1, Vam3, Vti1, Vam7), and of Sec18 starting from the coordinates of synaptobrevin-2, SNAP-25, syntaxin-1A, and NSF in the cryo-EM structure of the neuronal NSF/αSNAP/SNARE complex (PDB ID 3J96) (*Zhao et al., 2015*). For Sec18, the linker between the N and D1 domains was deleted from the generated homology model since there was no information about these linkers in this structure (PDB ID 3J96) of the neuronal 20S complex.

The MODELLER protocol consisted of an alignment step (python script file align.py) and a modeling step (python script file modeler-input.py). The script files are shown here for synaptobrevin (nyv1):

### align.py

```
from modeller import *
env = environ()
aln = alignment(env)
mdl = model(env, file='sb.pdb', model_segment=('FIRST:K','LAST:K'))
aln.append_model(mdl, align_codes='sbK', atom_files='sb.pdb')
aln.append(file='target_sequence.pir', align_codes='nyv1')
aln.salign(local_alignment = True, rr_file='${LIB}/blosum62.sim.mat',
gap_penalties_1d=(−600,–600),
output='',
align_block = 15, # no. of seqs. in first MSA
align_what='PROFILE',
alignment_type='PAIRWISE',
comparison_type='PSSM', # or 'MAT' (Caution: Method NOT benchmarked# for 'MAT')
similarity_flag = True, # The score matrix is not rescaled
substitution = True, # The BLOSUM62 substitution values are
# multiplied to the corr. coef.
#write_weights = True,
#output_weights_file='test.mtx', # optional, to write weight matrix
smooth_prof_weight = 10.0) # For mixing data with priors
aln.edit(edit_align_codes='nyv1', base_align_codes='rest',min_base_entries = 1,
overhang = 0)
aln.write(file='nyv1.ali', alignment_format='PIR')
aln.write(file='nyv1.pap', alignment_format='PAP')
```

### modeler-input.py

```
from modeller import *
from modeller.automodel import *
env = environ()
a = automodel(env, alnfile='nyv1.ali', knowns='sbK', sequence='nyv1', assess_methods=
(assess.DOPE, assess.GA341))
a.very_fast()
a.starting_model = 1
a.ending_model = 1
a.make()
```

The homology models of Nyv1, Vam3, Vti1, Vam7, and Sec18, together with the crystal structure of Sec17 (PDB ID 1QQE) (*Rice and Brunger, 1999*), were fit into the cryo-EM structure of the neuronal NSF/αSNAP/SNARE complex (PDB ID 3J96) (*Zhao et al., 2015*). The fit was performed by using the 'align' feature of PyMol to individually superimpose the coordinates of the vacuolar proteins with the corresponding coordinates of the neuronal proteins in the structure of the neuronal NSF/α SNAP/SNARE complex.

## Acknowledgements

We thank Gustav Lienhard, Charles Barlowe, Frederick Hughson, Michael Zick, Christian Ungermann, Jose Rizo, and Randy Schekman for insightful suggestions. This work was supported by grants R35GM118037 (to WW) and R37MH63105 (to ATB) from the NIH.

## Additional information

### Competing interests

Axel T Brunger: Reviewing editor, *eLife*. The other authors declare that no competing interests exist.

### Funding

| Funder | Grant reference number | Author |
|---|---|---|
| National Institutes of Health | R35GM118037 | William T Wickner |
| National Institutes of Health | R37MH63105 | Axel T Brunger |

The funders had no role in study design, data collection and interpretation, or the decision to submit the work for publication.

### Author contributions

Hongki Song, Conceptualization, Data curation, Software, Formal analysis, Validation, Investigation, Visualization, Methodology, Writing - review and editing; Thomas L Torng, Conceptualization, Data curation, Software, Formal analysis, Investigation, Visualization, Methodology, Writing - review and editing; Amy S Orr, Conceptualization, Data curation, Formal analysis, Investigation, Visualization, Methodology, Writing - review and editing; Axel T Brunger, Conceptualization, Resources, Data curation, Software, Formal analysis, Funding acquisition, Visualization, Project administration, Writing - review and editing; William T Wickner, Conceptualization, Resources, Supervision, Funding acquisition, Writing - original draft, Project administration, Writing - review and editing

### Author ORCIDs

Hongki Song (iD) https://orcid.org/0000-0002-3761-5434
Thomas L Torng (iD) http://orcid.org/0000-0003-2295-2777
Amy S Orr (iD) https://orcid.org/0000-0002-0266-0390
Axel T Brunger (iD) http://orcid.org/0000-0001-5121-2036
William T Wickner (iD) https://orcid.org/0000-0001-8431-0468

### Decision letter and Author response

Decision letter https://doi.org/10.7554/eLife.67578.sa1
Author response https://doi.org/10.7554/eLife.67578.sa2

## Additional files

### Supplementary files

- Transparent reporting form

### Data availability

All data generated or analyzed during this study are included in the manuscript and supporting files. Source data files have been provided for Figures 1, 2, 3 4, 5, and 6.

The following previously published datasets were used:

| Author(s) | Year | Dataset title | Dataset URL | Database and Identifier |
|---|---|---|---|---|
| Zhao M, Wu S, | 2015 | Structure of 20S supercomplex | https://www.rcsb.org/ | RCSB Protein Data |

| | | | |
|---|---|---|---|
| Cheng Y, Brunger AT | | determined by single particle cryoelectron microscopy (State I) | structure/3j96 | Bank, 3J96 |
| Rice LM, Brunger AT | 1999 | CRYSTAL STRUCTURE OF THE VESICULAR TRANSPORT PROTEIN SEC17 | https://www.rcsb.org/structure/1QQE | RCSB Protein Data Bank, 1QQE |

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
