## [Decision Letter]

**Acceptance summary:**

This is a very important paper that challenges the generally accepted dogma that full zippering of SNARE complexes is essential for intracellular membrane fusion. Previous work had already shown that C-terminal truncation of one SNARE arrested liposome fusion mediated by the yeast vacuolar SNARE complex and that Sec17/Sec18 could rescued fusion, but it was argued that such rescue could arise because Sec17/Sec18 restored C-terminal zippering. This paper now shows that Sec17/Sec18 rescue fusion even when three SNAREs are crippled -by truncation or mutation- to definitively prevent zippering, thus showing that Sec17/18 have a direct, positive role in membrane fusion.

**Decision letter after peer review:**

Thank you for submitting your article "Sec17/Sec18 can support membrane fusion without help from completion of SNARE zippering" for consideration by *eLife*. Your article has been reviewed by 3 peer reviewers, and the evaluation has been overseen by Josep Rizo as Reviewing Editor and Vivek Malhotra as the Senior Editor.

The reviewers have discussed their reviews with one another, and the Reviewing Editor has drafted this letter to help you prepare a revised submission.

Essential revisions:

1. The paper is difficult to read, particularly for non-experts in the field, and is very long, presenting the most dramatic results that support the main conclusion in Figure 10. The data in this figure are minimally dependent on the 51 data panels (97 including supplemental figures) preceding this figure. The reviewers wonder how many readers will make it that far and believe that a short, crisp manuscript would better convey the main message of the paper. One possibility is to reformat the paper to a Short Report including a small number of experiments. If the authors feel that such format is too constraining, they could keep the paper as a full Article but should strive to shorten it considerably. Note that some of the data are not necessary for the story and/or are not very compelling. For instance the FRET data of Figure 3, the HOPS/Sec17 binding data of Figure 4 and the studies with antibodies of Figure 9 could be part of a future paper devoted to more in depth mechanistic studies, and the titrations of Figure 7 could be included as supplementary material or could also be presented in such future study. In revising the manuscript, the authors should also consider the recommendations for the authors listed below. Note that these recommendations were extracted (unedited) from the separate reviews and some of them may be useful for a future paper(s) describing data that were removed from this manuscript.

2. The authors should discuss whether the mechanism underlying fusion in the in vitro assays presented in this paper could be operating in a cell under normal conditions or it only occurs when mutations are introduced into the SNAREs AND when Sec17/Sec18 are added above a threshold that might or might not be achieved in a cell (see point #1 of reviewer 3 below).

*Reviewer #1 (Recommendations for the authors):*

1. The authors should provide an explanation of why the observed FRET efficiency is so low even under the conditions under which SNARE complex assembly is most efficient and the donor and acceptor probes are expected to be very close. Is it due to low incorporation of the dyes on the proteins? It would be advisable to measure the FRET efficiency observed when the SNARE complex is assembled in solution, either with the proteins in the presence of detergent, or using only soluble SNARE domains. Does this lead to much more efficient FRET? If it does, why is SNARE complex formation on the liposomes much less efficient?

2. The concept of 'entropic cage' does not seem appropriate for the model that the authors are proposing, as favorable interactions at the C-terminus of the SNARE domains should be pretty much abolished by the mutations to Ser or Gly plus the truncations of two SNAREs. It seems very likely that any proximity between the C-termini of the two membrane-anchored SNAREs would arise from steric constraints caused by the bound Sec17 molecules rather than by entropic contributions. Perhaps the term 'steric cage' or simply 'cage' seem more appropriate.

3. Line 559: it seems likely that the persistence of fusion with the alanine mutations but not with the Ser mutations arises because there is still some hydrophobicity left, rather than because of higher helical propensity.

4. In lines 133-134, the authors state that αSNAP stabilizes SNARE complexes, citing Ma et al. 2016, but a more balanced view would be provided if the authors also cite another single molecule study that reached the opposite conclusion [Ryu et al., Science 347, 1485 (2015)].

5. The label 'No Sec17, Sec18 or ATP' at the top of Figure 8 is confusing, as there is Sec17 anchored on the liposomes in some of the experiments of the top panels.

6. Line 518: Qc3delta is not included in set 2 according to Figure 9.

7. Line 447: mM should be μM.

*Reviewer #2 (Recommendations for the authors):*

1. In Figure 1C, I wonder what the point is of showing "sticks" in addition to "cartoon". To me this makes the images look fuzzy, without adding any usable information.

2. In Figure 3, I'm not sure I understand why cis-SNARE complexes require Sec17 to complete the process of zippering? (And if they're not completely zippered, why does the stability of complex to exchange in the absence of Sec17 depend on the presence of the Qc-SNARE C-terminal layer residues [Figure 3F]?)

3. In Figure 3C, I was surprised that HOPS stimulation is independent of Ypt7. What do the authors make of this?

4. In Figure 7, is there evidence of cooperativity with respect to Sec17? Mightn't it be interesting to investigate this and perhaps determine a Hill coefficient?

5. Also in Figure 7, why does Sec18 still have a dramatic stimulatory effect (compare 1000 nM curves in panels G and H) when Sec17 contains the LALA mutation (which ought to prevent Sec18 binding)?

6. Figure 9A Set 4, indicating that Qc alone is sufficient to confer resistance to αVps33, would seem to be quite informative. Would it be interesting to repeat the experiment with Qb in place of Qc?

7. What might the partial resistance to αSec18 in Figure 9, Sets 8 and 9 mean?

8. The data in Figure 10 doesn't seem to agree with the data in Figure 10 —figure supplement 1.

*Reviewer #3 (Recommendations for the authors):*

1. It is striking that, in the presence of Sec17, Sec18, and HOPS, SNAREs are capable of driving efficient fusion even when several C-terminal layers are deleted. While the data are overall convincing, I am wondering if the Sec17-dependent activation is utilized in the cell with wild-type SNAREs. In other words, is such a mechanism also relevant beyond the engineered system described here? To answer this question, the authors would need to isolate Sec17 or Sec18 mutants that are only defective in stimulating fusion while remaining fully functional in cis-SNARE disassembly. Then they could test if such mutants are defective in driving fusion in vivo.

2. The authors stated that the presence of SNAP-25 linker interferes with the binding with the other two α-SNAP molecules. Does this mean the mechanism proposed in this work is specific to yeast vacuole fusion rather than a general principle?

[Editors' note: further revisions were suggested prior to acceptance, as described below.]

Thank you for resubmitting your work entitled "Sec17/Sec18 can support membrane fusion without help from completion of SNARE zippering" for further consideration by *eLife*. Your revised article has been evaluated by Vivek Malhotra (Senior Editor), Josep Rizo as Reviewing Editor and one of the referees who reviewed the original version.

The manuscript has been improved but there are some remaining concerns that need to be addressed, as outlined below:

1. A key issue that needs to be clarified is why does the zippering of cis-SNARE complexes depend on Sec17 (Figure 2)? In these experiments, the R- and Qa-SNARE are embedded in the same liposomes, and the Qb- and Qc-SNAREs lack transmembrane domains. The failure of such complexes to fully zipper makes no sense unless something prevents such zippering and the natural culprit is HOPS. The authors should briefly discuss the mechanistic implications of these findings. For example, might HOPS have a proofreading function, in which it somehow prevents cis-SNARE complexes from fully zippering (but perhaps not trans-SNARE complexes)? In any case, the authors should clearly point out from the beginning of the description of the zippering experiments of Figure 2 that they involve cis-SNARE complexes. They should also explain the relevance of cis-SNARE zippering assays to a manuscript that otherwise deals with trans-SNARE zippering/fusion.

2. The use of lines and sticks in Figure 1C is visually uninterpretable and therefore unhelpful. The authors are advised to remove them although this is of course up to them.

---

## [Author Response]

Essential revisions:1. The paper is difficult to read, particularly for non-experts in the field, and is very long, presenting the most dramatic results that support the main conclusion in Figure 10. The data in this figure are minimally dependent on the 51 data panels (97 including supplemental figures) preceding this figure. The reviewers wonder how many readers will make it that far and believe that a short, crisp manuscript would better convey the main message of the paper. One possibility is to reformat the paper to a Short Report including a small number of experiments. If the authors feel that such format is too constraining, they could keep the paper as a full Article but should strive to shorten it considerably. Note that some of the data are not necessary for the story and/or are not very compelling. For instance the FRET data of Figure 3, the HOPS/Sec17 binding data of Figure 4 and the studies with antibodies of Figure 9 could be part of a future paper devoted to more in depth mechanistic studies, and the titrations of Figure 7 could be included as supplementary material or could also be presented in such future study. In revising the manuscript, the authors should also consider the recommendations for the authors listed below. Note that these recommendations were extracted (unedited) from the separate reviews and some of them may be useful for a future paper(s) describing data that were removed from this manuscript.

We have taken this seriously, and now have 5 data figures (Figures 2-6) instead of the 9 in the original submission. For this reason, and since the Introduction and Results are more succinct, the reader won't "burn out" before getting to the "triply-crippled Q's" Figure. The 5 data figures are essential to develop the system and show its characteristics, and the supplementary figures provide depth to our consideration of the Figures. For a paper which challenges a main shibboleth of the field, we feel it's important to develop the system fully, showing the full impact and ramifications of crippling even one Q, rather than just cherry-pick one culminating figure.

2. The authors should discuss whether the mechanism underlying fusion in the in vitro assays presented in this paper could be operating in a cell under normal conditions or it only occurs when mutations are introduced into the SNAREs AND when Sec17/Sec18 are added above a threshold that might or might not be achieved in a cell (see major point #1 of reviewer 3 below).Reviewer #1 (Recommendations for the authors):1. The authors should provide an explanation of why the observed FRET efficiency is so low even under the conditions under which SNARE complex assembly is most efficient and the donor and acceptor probes are expected to be very close. Is it due to low incorporation of the dyes on the proteins? It would be advisable to measure the FRET efficiency observed when the SNARE complex is assembled in solution, either with the proteins in the presence of detergent, or using only soluble SNARE domains. Does this lead to much more efficient FRET? If it does, why is SNARE complex formation on the liposomes much less efficient?

We emphasize that these are ensemble FRET experiments rather than single-molecule FRET experiments. The average FRET efficiency is modest because its denominator includes all the fluorescent Qb and Qc in a bulk reaction, even though only a small per cent enter HOPS-dependent SNARE complex. We note this now directly in the text.

2. The concept of 'entropic cage' does not seem appropriate for the model that the authors are proposing, as favorable interactions at the C-terminus of the SNARE domains should be pretty much abolished by the mutations to Ser or Gly plus the truncations of two SNAREs. It seems very likely that any proximity between the C-termini of the two membrane-anchored SNAREs would arise from steric constraints caused by the bound Sec17 molecules rather than by entropic contributions. Perhaps the term 'steric cage' or simply 'cage' seem more appropriate.

Thank you, we have removed this term and now describe our model in terms of an "environment" that Sec17 creates.

3. Line 559: it seems likely that the persistence of fusion with the alanine mutations but not with the Ser mutations arises because there is still some hydrophobicity left, rather than because of higher helical propensity.

A good point, which we have now added. Thanks.

4. In lines 133-134, the authors state that αSNAP stabilizes SNARE complexes, citing Ma et al. 2016, but a more balanced view would be provided if the authors also cite another single molecule study that reached the opposite conclusion [Ryu et al., Science 347, 1485 (2015)].

Thanks, we've now added this.

5. The label 'No Sec17, Sec18 or ATP' at the top of Figure 8 is confusing, as there is Sec17 anchored on the liposomes in some of the experiments of the top panels.

Thanks, our mistake! We have removed "Sec17" from these labels; as you note, what differentiates these rows of panels is the presence or absence of Sec18 and of ATP or its analogs.

6. Line 518: Qc3delta is not included in set 2 according to Figure 9.

Qc3delta is not included initially, but is added along with Sec17, Sec18, and ATPgammaS at 30min., 1min after the control IgG, anti-Vps33, or anti-Sec18 were added. Each of the 8 sets has all the same components, but differ in which are added from the start and which are only added after the antibodies; we've now emphasized this in the text.

7. Line 447: mM should be μM.

Thank you.

Reviewer #2 (Recommendations for the authors):1. In Figure 1C, I wonder what the point is of showing "sticks" in addition to "cartoon". To me this makes the images look fuzzy, without adding any usable information.

We included the sidechains to emphasize that these molecules are extensively interacting rather than just being near each other.

2. In Figure 3, I'm not sure I understand why cis-SNARE complexes require Sec17 to complete the process of zippering? (And if they're not completely zippered, why does the stability of complex to exchange in the absence of Sec17 depend on the presence of the Qc-SNARE C-terminal layer residues [Figure 3F]?)

Without Sec17, these complexes apparently don't zipper all the way most of the time; they are in a metastable partially-zippered state. Fusion without Sec17 is slow because zippering is slow/infrequent; adding Sec17 enhances zippering and enhances fusion. When Qc lacks the +4 to +8 layers, it's perhaps zippered even less, and is seen to be labile to exchange.

3. In Figure 3C, I was surprised that HOPS stimulation is independent of Ypt7. What do the authors make of this?

As we've noted in earlier papers (Stroupe et al., 2006), HOPS has direct affinity for SNAREs, and a strict requirement for Ypt7 is only seen at low SNARE levels (Zick and Wickner, 2016). The FRET signals of SNARE complexes, by the assay in Figure 2, would have been too low had we employed the very low SNARE levels which confer strict Ypt7-dependence.

4. In Figure 7, is there evidence of cooperativity with respect to Sec17? Mightn't it be interesting to investigate this and perhaps determine a Hill coefficient?

We agree, but a more detailed study will be needed to determine a Hill coefficient.

5. Also in Figure 7, why does Sec18 still have a dramatic stimulatory effect (compare 1000 nM curves in panels G and H) when Sec17 contains the LALA mutation (which ought to prevent Sec18 binding)?

The LALA mutant blocks Sec17 stimulation of Sec18-mediated 4-SNARE complex disassembly, but hasn't been shown to block all Sec17:Sec18 binding.

6. Figure 9A Set 4, indicating that Qc alone is sufficient to confer resistance to αVps33, would seem to be quite informative. Would it be interesting to repeat the experiment with Qb in place of Qc?

This is just what was done in Set 2.

7. What might the partial resistance to αSec18 in Figure 9, Sets 8 and 9 mean?

We can only speculate about the reasons. For example, the partial resistance would be consistent with HOPS:4-SNARE assembly creating a very favorable binding site for Sec18, so that the SNAREs binding Sec18 yields an antibody-resistant complex about as fast as other Sec18s are captured in solution and inactivated by their antibody. Perhaps when Sec18 is added from the start without Sec17, as in Set 6, it binds to non-functional SNARE subcomplexes.

8. The data in Figure 10 doesn't seem to agree with the data in Figure 10 —figure supplement 1.

We made a labeling error and have fixed it.

Reviewer #3 (Recommendations for the authors):1. It is striking that, in the presence of Sec17, Sec18, and HOPS, SNAREs are capable of driving efficient fusion even when several C-terminal layers are deleted. While the data are overall convincing, I am wondering if the Sec17-dependent activation is utilized in the cell with wild-type SNAREs. In other words, is such a mechanism also relevant beyond the engineered system described here? To answer this question, the authors would need to isolate Sec17 or Sec18 mutants that are only defective in stimulating fusion while remaining fully functional in cis-SNARE disassembly. Then they could test if such mutants are defective in driving fusion in vivo.

This is one of our long-term plans! For now though, we do point out that Schwartz et al., (2017) showed that the vacuole fragmentation phenotype from Qc3delta is cured by Sec17 overexpression. We've also added to the text that our concentrations of Sec17 and Sec18 are in the same range as published in vivo concentrations; the intracellular concentrations of Sec17 and Sec18 are 150 to 1100nM and 250-760nM, respectively (Ho et al., 2018, now added to our references).

2. The authors stated that the presence of SNAP-25 linker interferes with the binding with the other two α-SNAP molecules. Does this mean the mechanism proposed in this work is specific to yeast vacuole fusion rather than a general principle?

A more thorough investigation of other fusion systems will be needed to evaluate the generality of Sec17/Sec18 direct promotion of fusion. Many other Qb and Qc SNAREs are not joined as in SNAP-25, but we don't know whether or not this will correlate with whether Sec17/Sec18 stimulate fusion.

We note that our model doesn't depend on the precise stoichiometry of Sec17:SNAREs. CryoEM structures of NSF/SNAP complexes suggest 2:1, 3:1, or 4: stoichiometries.

[Editors' note: further revisions were suggested prior to acceptance, as described below.]The manuscript has been improved but there are some remaining concerns that need to be addressed, as outlined below:1. A key issue that needs to be clarified is why does the zippering of cis-SNARE complexes depend on Sec17 (Figure 2)? In these experiments, the R- and Qa-SNARE are embedded in the same liposomes, and the Qb- and Qc-SNAREs lack transmembrane domains. The failure of such complexes to fully zipper makes no sense unless something prevents such zippering and the natural culprit is HOPS. The authors should briefly discuss the mechanistic implications of these findings. For example, might HOPS have a proofreading function, in which it somehow prevents cis-SNARE complexes from fully zippering (but perhaps not trans-SNARE complexes)? In any case, the authors should clearly point out from the beginning of the description of the zippering experiments of Figure 2 that they involve cis-SNARE complexes. They should also explain the relevance of cis-SNARE zippering assays to a manuscript that otherwise deals with trans-SNARE zippering/fusion.

Why does zippering need Sec17? We agree that "...the natural culprit is HOPS", and have added emphasis to this for clarity, now explicitly noting that "… the enhanced FRET between the SNAREs in the presence of HOPS and Sec17 (bar 9 [of Figure 2]) is not seen without HOPS (bar 8) or Sec17 (bar 10)." We hadn't stated this clearly enough before.

We now point out explicitly that the model we're examining in Figure 2 is of *cis*-SNARE complexes. We also more clearly cast the study in Figure 2 as one way to examine the interactions of HOPS, Sec17, and the SNAREs, a preamble to functional studies presented in the later figures. This is now pointed out both right before Figure 2 and in the Discussion.

We now more cautiously refer throughout to Sec17/HOPS induced conformational change in the SNAREs, reflected by altered FRET, rather than just "zippering". We discuss why zippering is a reasonable, even likely, explanation for this conformational change.

We too yearn for a physical assay of the completion of zippering of SNAREs in trans, but this is not yet feasible: trans-SNARE complexes are far less abundant than cis-complexes, engaging no more than a few per cent of the SNAREs (Figure 2, Collins and Wickner, 2007). Furthermore, introducing two somewhat bulky fluorophores in the crowded environment at the C-termini of SNARE domains of trans-complex with HOPS, Sec17, and two tightly apposed membranes could well induce artifacts.

2. The use of lines and sticks in Figure 1C is visually uninterpretable and therefore unhelpful. The authors are advised to remove them although this is of course up to them.

We're removed the "lines and sticks" from Figure 1C, as requested.